# MAXIMUM NEXT-STATE ENTROPY FOR EFFICIENT REINFORCEMENT LEARNING

## ABSTRACT

Maximum entropy algorithms have demonstrated significant progress in Reinforcement Learning (RL), which offers additional guidance in the form of entropy, which is particularly beneficial in tasks with sparse rewards. Nevertheless, current approaches grounded in policy entropy encourage the agent to explore diverse actions, yet they do not directly help the agent explore diverse states. In this study, we theoretically reveal the challenge of optimizing the next-state entropy of the agent. To address this limitation, we introduce **M**aximum **N**ext-**S**tate **E**ntropy (**MNSE**), a novel method that maximizes next-state entropy through an action mapping layer following the inner policy. We provide a theoretical analysis demonstrating that MNSE can maximize next-state entropy by optimizing the action entropy of the inner policy. We conduct extensive experiments on various continuous control tasks and show that MNSE can significantly improve the exploration capability of RL algorithms.

## 1 INTRODUCTION

Maximum entropy RL algorithms have demonstrated remarkable performance across various domains, including games (Gao et al., 2018), robotic control (Haarnoja et al., 2018a;b), and autonomous navigation (Sun et al., 2022). The maximum entropy framework enhances policy exploration and robustness by optimizing both reward and policy entropy simultaneously (Ziebart et al., 2008; 2010). Recent advancements adapt the temperature dynamically (Hu et al., 2021)to improve the trade-off between reward and entropy. Meanwhile, Deep Soft Policy Gradient (DSPG) Shi et al. (2019) integrates soft policy gradients with the soft Bellman equation to address stability issues in off-policy learning.

However, maximizing policy entropy may not directly help the agent explore diverse states due to redundancy in the action space. For example, in real-world tasks, actuators often exhibit saturation and dead zone issues due to design redundancies or equipment aging (Galuppini et al., 2018; Bai, 2002). This leads to a redundant action space, where multiple actions can result in the same state. If we cannot deal with this issue, the nonlinear actuators can reduce control accuracy, thereby significant degrading the control performance.

To help the agent directly explore diverse states, current researchers propose novelty-based exploration methods, such as random network distillation (Burda et al., 2018) and pseudo-count techniques (Lobel et al., 2023; Machado et al., 2020). These methods aim to drive agents toward discovering more diverse states by directly evaluating the novelty of a state and incorporating it as an exploration bonus into the extrinsic reward. Current novelty-based and pseudo-count exploration methods have achieved considerable success across various domains. However, it is unclear whether state diversity can provably benefit maximum entropy RL and how to bridge the gap under theoretical guarantees, which naturally leads to the following question:

*How can we maximize the next-state entropy of the RL agent in a principled way?*

To answer this question, we start with the analysis of next-state entropy. Intuitively, if the policy is non-redundant, we can optimize the next-state entropy by optimizing policy entropy. However, the condition of non-redundant policy is hard to meet in the stochastic transition case. To solve this issue, we first introduce the inner policy and reversible action mapping layer. Then, we rigorously derive the gap between the next-state entropy and the policy entropy of the inner policy. We

demonstrate that by optimizing both the action mapping layer and inverse dynamics model, we can maximize the next-state entropy by optimizing the inner policy entropy. In practice, our method also achieves superior performance. We conduct extensive experiments on various environments, including MuJoCo (Todorov et al., 2012) and Meta-World (Yu et al., 2020). Results show that our algorithm performs better than baselines.

## 2 RELATED WORK

**Maximum Entropy Reinforcement Learning** has been widely adopted for improving policy exploration and robustness in RL. Early work introduced Soft Actor-Critic (SAC) (Haarnoja et al., 2018c;d), an off-policy actor-critic algorithm that formalizes MaxEnt RL by balancing the goals of maximizing expected return and policy entropy. Deep Soft Policy Gradient (DSPG) by (Shi et al., 2019) integrates soft policy gradients with the soft Bellman equation to address stability issues in off-policy learning. Count-Based Soft Q-Learning (CBSQL) by (Hu et al., 2021) adapts the temperature dynamically to improve the trade-off between reward and entropy. Additionally, Han & Sung (2021) propose a max-min entropy framework to improve exploration in model-free learning by promoting low-entropy state visitation.

**Action Representation.** Extensive efforts have been made to effectively represent actions within large action spaces. Zahavy et al. (2018b) propose a method that directly identifies redundant or irrelevant actions using external elimination signals provided by the environment, removing them from the sampling process in text-based games. Tennenholtz & Mannor (2019) adopt a negative sampling procedure, leveraging expert demonstrations to better understand the action space. However, valuable prior information is often scarce and expensive, limiting the scalability of these approaches. Chandak et al. (2019) demonstrates how to learn and utilize action representations without relying on prior knowledge by embedding them within the policy structure to train agents effectively. Similarly, Metz et al. (2017) introduces a novel approach to discretize high-dimensional continuous action spaces by sequentially combining one-dimensional discrete actions.

**Exploration** is a cornerstone of reinforcement learning, with various strategies enhancing agents' ability to learn from complex environments. For example, the Go-Explore strategy (Ecoffet et al., 2019) advocates for a phased approach to overcoming challenging exploration dilemmas. Count-based methods (Bellemare et al., 2016) capitalize on environmental novelty by employing pseudo-counts. Disagreement-based exploration (Pathak et al., 2019) harnesses the variance in model predictions to propel the agent toward exploration. Curiosity-driven exploration mechanisms, such as ICM (Pathak et al., 2017), utilize prediction errors to incentivize exploration. RND (Burda et al., 2018) employs a novel neural network to generate intrinsic rewards based on the prediction error of environmental dynamics, driving the agent towards unexplored territories. NGU (Badia et al., 2020) integrates intrinsic motivation with an episodic memory mechanism to encourage the revisitation of novel states, promoting long-term exploration.

**State Entropy Maximization** aims to learn a reward-free policy in which state visitations are uniformly distributed across the state space, thus promoting robust policy initialization and efficient adaptation. Additionally, when task rewards are available, incorporating state entropy as an intrinsic reward has proven to be an effective approach for enhancing exploration. Lee et al. (2019) propose optimizing the state marginal distribution to align with a target distribution, effectively enhancing exploration. Building on this idea, Islam et al. (2019) introduce entropy regularization based on the marginal state distribution, achieving superior state space coverage in complex domains. Further advancements include the work of Guo et al. (2021), who incorporate geometry-aware Shannon entropy of state visitations in both discrete and continuous domains, framing exploration as a computationally tractable problem. Additionally, Hazan et al. (2019) provide a provably efficient algorithm for state entropy maximization, leveraging a black-box planning oracle. Expanding on these methods, Liu & Abbeel (2021) maximize a particle-based entropy in an abstract representation space, demonstrating human-level performance in navigating complex environments.

## 3 BACKGROUND

**Reinforcement Learning.** We consider the Markov Decision Processes (MDPs) as the model process, defined by the tuple $(\mathcal{S}, \mathcal{A}, \mathcal{P}, r, \gamma)$, where $\mathcal{S}$ is a state space, $\mathcal{A}$ is an action space, $\gamma \in [0, 1)$

is the dicount factor and $\mathcal{P} : \mathcal{S} \times \mathcal{A} \to Dist(\mathcal{S}), r : \mathcal{S} \to [r_{\min}, r_{\max}]$ are the transition function and reward function, respectively. We assume a fixed distribution $\mu_0$ as the initial state distribution. The goal of an RL agent is to learn a policy $\pi(a \mid s)$ under dataset $\mathcal{D}$, which maximizes the expectation of a discounted cumulative reward: $J(\pi) = \mathbb{E}_{\mu_0, \pi} \left[ \sum_{t=0}^{\infty} \gamma^t r_t \right]$. For any policy $\pi$, the corresponding state-action value function is $Q^\pi(s, a) = \mathbb{E}[\sum_{k=0}^{\infty} \gamma^k r_{t+k} | S_t = s, A_t = a, \pi]$.

**Maximum Policy Entropy.** Different with standard reinforcement learning, maximum policy entropy reinforcement learning aims to augment the objective with the expected entropy of the policy:

$$J(\pi) = \mathbb{E}_{\mu_0, \pi} \left[ \sum_{t=0}^{\infty} \gamma^t \left( r_t + \alpha \mathcal{H}(\pi(\cdot \mid s_t))) \right) \right], \tag{1}$$

where $\alpha$ is a temperature parameter, determining the relative importance of the entropy term against the reward, and thus controls the stochasticity of the optimal policy.

# 4 ANALYSIS OF NEXT-STATE ENTROPY

*How do we maximize the state entropy of the agent?* Current approaches have been to encourage exploration by adding bonus rewards related to the new state (Burda et al., 2018; Badia et al., 2020; Zhang et al., 2021). Adding an exploration bonus has achieved considerable success, while theoretical analysis of the state entropy has not been explored well. To bridge this gap, in this section, we theoretically analyze the optimization objective of the next-state entropy. Firstly, we define the next-state entropy as $\mathcal{H}(S_{t+1} \mid S_t = s, \pi)$, which represents the entropy of the next state after executing policy $\pi$ in state $s$.

**Definition 4.1** (Next-State Entropy). *We define the next-state entropy under policy $\pi$ following state $s$ by*

$$\mathcal{H}(S_{t+1} \mid S_t = s, \pi) = -\mathbb{E}_{a \sim \pi(\cdot \mid s)} \mathbb{E}_{s' \sim P(\cdot \mid s, a)} \log \left[ P^\pi(s' \mid s) \right] \tag{2}$$

where $P^\pi(s' \mid s) = \int_{a \in \mathcal{A}} \pi(a \mid s) P(s' \mid s, a)$. The **next-state entropy**, as defined above, measures the diversity of the subsequent states under the policy $\pi$. In classical entropy-regularized reinforcement learning, the **policy entropy** is often used to encourage diversity in the actions taken. However, these two concepts are not generally equivalent.

In this section, we will illustrate the discrepancy between policy entropy and next-state entropy during policy updates in an illustrative example (Section 4.1) and then reveal their relationship under both deterministic (Section 4.2) and stochastic transitions (Section 4.3) with theoretical analysis.

## 4.1 TOY EXAMPLE

The maximum entropy RL framework is often credited with improving exploration efficiency and promoting more diverse state visitation, particularly in sparse reward settings. However, through the following illustrative example, we demonstrate that optimizing policy entropy alone can be inefficient in certain cases, especially when there is redundancy in the action space.

Consider a one-step MDP with a deterministic transition, where $s_{t+1} = \max(a_t, 0)$, which means actions less than zero are redundant. We assume an initial policy $\pi_{policy}$ [1] is a Gaussian policy with a mean less than zero: $\pi_{policy}(a) = \frac{1}{\sqrt{2\pi}\sigma} \exp\left(-\frac{(a-\mu)^2}{2\sigma^2}\right), \mu < 0$. The corresponding policy entropy is given by:

$$\mathcal{H}(\pi_{policy}(\cdot \mid s_t)) = \log(\sqrt{2\pi e}\sigma), \tag{3}$$

which is independent of the mean $\mu$. As shown in the Figure 1 (Left), updating the policy by maximizing policy entropy alone leads to increased variance $\sigma$. However, since the mean of the Gaussian distribution remains unchanged, there is still more than a 50% probability (the gray area) that actions sampled from the updated policy will be less than zero, leading to the same next state.

---

[1] In Section 4.1, to avoid confusion between the policy $\pi$ and the mathematical constant $\pi$, we denote the policy by $\pi_{policy}$.

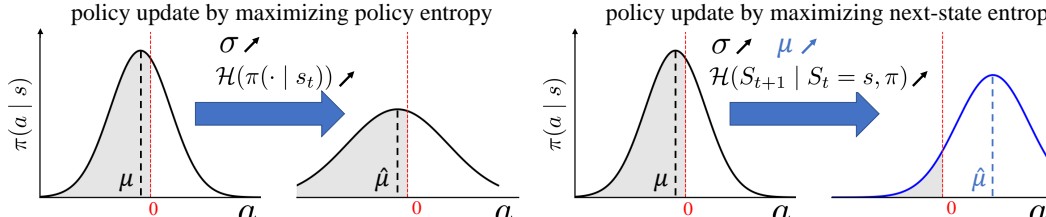

Figure 1: Probability distribution of policy $\pi$. For a one-step MDP with a deterministic transition $s_{t+1} = \max(a_t, 0)$, the gray area shows the probability of actions leading to the same next state. **Left**: Policy update under maximum policy entropy by increasing variance $\sigma$. **Right**: Policy update under maximum next-state entropy by increasing both variance $\sigma$ and mean $\mu$.

In contrast, the next-state entropy can be mathematically expressed as:

$$\mathcal{H}\left(S_{t+1} \mid S_t = s, \pi_{policy}\right) = \underbrace{\log(\sqrt{2\pi e}\sigma)}_{\text{policy entropy}} + \left\{ \Phi(\frac{\mu}{\sigma})\log\sigma - \int_{-\infty}^{-\frac{\mu}{\sigma}} -\varphi(z)\log\varphi(z)dz \right\}, \quad (4)$$

where $\varphi(z) = \frac{e^{-z^2/2}}{\sqrt{2\pi}}$ is the probability density function of the standard normal distribution, and $\Phi(x) = \int_{-\infty}^{x} \varphi(z)dz$ is its cumulative distribution function. The Equation 4 highlights that next-state entropy is influenced not only by the policy entropy but also by an additional term associated with $\mu$. Notably, when we assume $\sigma > 1$ in the toy example, the variable $\mu$ positively correlates with next-state entropy. A detailed proof and numerical analysis are provided in Appendix A. As illustrated in the Figure 1 (Right), policy updates driven by next-state entropy increase both the variance $\sigma$ and the mean $\mu$, significantly reducing the probability of sampling actions less than zero, which decreases redundancy in the next states and enhance state diversity.

## 4.2 DETERMINISTIC CASE

Section 4.1 demonstrates the superiority of next-state entropy over policy entropy in terms of exploration. To better understand the relationship between next-state entropy and policy entropy, we start with an MDP with deterministic transitions. As discussed by Baram et al. (2021), the following corollary provides insights into the equivalent relationship between next-state entropy and policy entropy under deterministic case:

**Corollary 4.2.** *Let $M$ be a deterministic MDP with a transition function $T : \mathcal{S} \times \mathcal{A} \to \mathcal{S}$. If $T(s, a) \neq T(s, a'), \forall s \in \mathcal{S}, \forall a, a' \in \mathcal{A}$, then*

$$\mathcal{H}(S_{t+1} \mid S_t = s, \pi) = \mathcal{H}(\pi(\cdot \mid s_t)) \quad , \quad \forall s \in S$$

This corollary indicates that, in the deterministic case, if there are no redundant actions in the action space $\mathcal{A}$, next-state entropy and policy entropy are equivalent. Therefore, recent studies (Zahavy et al., 2018a; Tennenholtz & Mannor, 2019; Zhong et al., 2024) have focused on eliminating redundant actions in the action space $\mathcal{A}$ and significantly enhance the exploration and performance in various domains.

## 4.3 STOCHASTIC CASE

Similarly, in the stochastic case, we can define a non-redundant policy and reveal the relationship between next-state entropy and policy entropy. Firstly, we define the non-redundant policy as follows:

**Definition 4.3** (Non-Redundant Policy). *Given a stochastic MDP with a transition dynamics $P$, $\pi$ is a non-redundant policy if*

$$\forall s, s' \in \mathcal{S}, \quad \forall a_i, a_j \in \{a \in \mathcal{A} \mid \pi(a \mid s) > 0\}, \quad P(s' \mid s, a_i) * P(s' \mid s, a_j) = 0.$$

In the stochastic case, we define $\pi$ as a non-redundant policy only if there is an absolutely non-intersection between probability distribution $P(s' \mid s, a)$ of possible actions sampling from $\pi$. In other words, for any reachable next state $s'$, only one action can lead to it under a non-redundant policy. Under the above strong assumption on policy $\pi$, we can reveal the relationship between next-state entropy and policy entropy in the stochastic case:

**Theorem 4.4.** *If $\pi$ is a non-redundant policy, then*

$$\mathcal{H}(S_{t+1} \mid S_t = s, \pi) = \mathcal{H}(\pi(\cdot \mid s)) + \mathcal{H}_{\text{model}}, \tag{5}$$

*where $\mathcal{H}_{\text{model}} = \mathbb{E}_{a \sim \pi(\cdot \mid s)} \mathcal{H}(S_{t+1} \mid S_t = s, a)$ is the entropy of the dynamics model.*

*Proof.* Please refer to Appendix B for the detailed proof. $\square$

The entropy of the dynamics model $\mathcal{H}_{\text{model}}$ represents an inherent property of the system. Suppose we treat this term as a constant $c$. In that case, the above equation reveals that in the case of stochastic transitions, as long as the policy is non-redundant, next-state entropy is equivalent to policy entropy.

## 5 METHOD

Based on the analysis in Section 4, we can directly optimize state entropy by maximizing the entropy of the non-redundant policy. However, constructing a non-redundant policy in the stochastic transition setting is extremely challenging since we must strictly satisfy the Definition 4.3. To solve this issue, we decompose the overall policy into an inner policy $\pi_i$ and a parameterized, reversible action mapping layer $f$. Based on this framework, we rigorously derive the gap between the next-state entropy of the overall policy and the policy entropy of the inner policy. Meanwhile, we demonstrate that by optimizing both the action mapping layer and an inverse dynamics model to minimize the gap term, the inner policy's entropy will be equivalent to the next-state entropy of the overall policy.

### 5.1 ACTION MAPPING

We consider a new action space $\mathcal{E}$, where action $a \in \mathcal{A}$ is a function of the inner action $e \in \mathcal{E}$:

$$a = f(e; \theta), \quad \forall e \in \mathcal{E}, \tag{6}$$

where $f$ is the action mapping parameterized by $\theta$ and it has an invertible function:

$$e = f^{-1}(a; \theta), \quad \forall a \in \mathcal{A}. \tag{7}$$

Subsequently, we define policy based on the action mapping as follows: for a given state $s$, the inner policy $\pi_i(\cdot \mid s)$ outputs the action $e$ in new action space $\mathcal{E}$ and then $f$ transforms $e$ back to the original action space $\mathcal{A}$. For the policy $\pi$, we have the following conclusion:

$$\pi(a \mid s) = \pi_i(e \mid s) * \left| \frac{\partial f^{-1}(a; \theta)}{\partial a} \right| = \pi_i(e \mid s) * \left| \frac{\partial f(e; \theta)}{\partial e} \right|^{-1}, \tag{8}$$

where $\left| \frac{\partial f(e; \theta)}{\partial e} \right|$ is the Jacobian determinant of the function $f$. We denote the entropy of the internal policy as $\mathcal{H}(\pi_i(\cdot \mid s))$ and denote the entropy of the state as $\mathcal{H}(S_{t+1} \mid S_t = s, \pi)$. Following the above policy framework and notations, we can derive the entropy of the state as follows:

**Theorem 5.1.** *For any inner policy $e \sim \pi_i(\cdot \mid s)$ and invertible action mapping layer $a = f(e; \theta)$, we have*

$$\mathcal{H}(S_{t+1} \mid S_t = s, \pi) = \mathcal{H}(\pi_i(\cdot \mid s)) + \underbrace{\mathbb{E}_{a \sim \pi(\cdot \mid s)} \mathbb{E}_{s' \sim p(\cdot \mid s, a)} [\log [p_{\text{inv}}(e \mid s, s')]]}_{\text{Gap Term}} + \mathcal{H}_{\text{model}}$$

*where $\mathcal{H}_{\text{model}} = \mathbb{E}_{e \sim \pi_i(\cdot \mid s)} [\mathcal{H}(S_{t+1} \mid S_t = s, e)]$ is the entropy of the dynamics model, $p_{\text{inv}}(e \mid s, s')$ is the inverse dynamic of inner policy, which is a function that predicts the inner action $e$ required to transition from a current state $s$ to a next state $s'$.*

---

**Algorithm 1** Maximum Next-State Entropy RL

---

1: **Inputs:** Initialize inner policy $\pi_i^{\psi}$, inverse dynamics model $p_{\text{inv}}^{\phi}$, action mapping layer $f_{\theta}$.
2: **for** iteration $t = 0, 1, 2, ...$ **do**
3:    Observe state $s$ and select internal action $e \sim \pi_{\psi}^i(\cdot \mid s)$
4:    Execute action $a = f_{\theta}(e)$ in the environment
5:    Obtain next state $s'$, reward $r$, and done signal $d$
6:    Store $(s, e, a, r, s', d)$ in replay buffer $\mathcal{D}$
7:    **if** it's time to update **then**
8:       Update inverse dynamics model parameters $\phi$ based on Equation 11
9:       Update the action mapping layer parameters $\theta$ based on Theorem 5.2
10:      Update innner policy $\pi_i^{\psi}$ based on the standard maximum policy entorpy RL algorithms
11:   **end if**
12: **end for**
13: **Return:** Policy parameters $\psi$ and action mapping layer parameters $\theta$

---

*Proof.* Please refer to Appendix C for the detailed proof. $\qquad\square$

Theorem 5.1 suggests that we do not require $\pi_i$ is the non-redundant policy. In addition, it is noteworthy that the entropy of the dynamics model $\mathcal{H}_{\text{model}}$ is an inherent characteristic of the system. We can regard this term as a constant $c$. Therefore, if we can minimize the gap term, we can maximize the next-state entropy by optimizing the policy entropy of the inner policy $\pi_i$.

### 5.2 MAXIMIZE NEXT-STATE ENTROPY

Based on the analysis in Theorem 5.1, we can maximize next-state entropy as follows:

$$J_{\text{MNSE}}(\pi) = \mathop{\mathbb{E}}_{(s_t, a_t, s_{t+1}) \sim \pi} \left[ \sum_{t=0}^{\infty} \gamma^t \left( r_t + \alpha \mathcal{H}(S_{t+1} \mid S_t = s_t, \pi) \right) \right]$$

$$= \mathop{\mathbb{E}}_{(s_t, a_t, s_{t+1}) \sim \pi} \left[ \sum_{t=0}^{\infty} \gamma^t \left( r_t + \alpha \underbrace{\mathcal{H}(\pi_i^{\psi}(\cdot \mid s_t))}_{\text{Entorpy of Inner Policy}} \right. \right. \tag{9}$$

$$\left. \left. + \alpha \underbrace{\mathop{\mathbb{E}}_{a \sim \pi(\cdot \mid s_t)} \mathop{\mathbb{E}}_{s_{t+1} \sim P(\cdot \mid s, a)} \left[ \log \left[ p_{\text{inv}}^{\phi}(e = f^{-1}(a_t; \theta) \mid s_t, s_{t+1}) \right] \right]}_{\text{Gap Term}} + c \right) \right],$$

where the inverse dynamics $p_{\text{inv}}^{\phi}$ is parameterized with $\phi$, and the inner policy $\pi_i^{\psi}$ is parameterized with $\psi$. It is noteworthy that $\pi(\cdot \mid s)$ in the above equation implicitly includes the inner policy $\pi_i^{\psi}$ and the action mapping layer $f_{\theta}$ defined in the Equation 6. Therefore, we need to optimize these three parameters $\phi, \psi, \theta$ simultaneously.

Based on the Equation 9, we can conclude that maximizing $J_{\text{MNSE}}(\pi)$ is equivalent to maximizing the entropy of the inner policy $\pi_i^{\psi}$ if and only if the *Gap Term* is zero. Further, with an appropriate model to estimate the inverse dynamics, since $\log p_{\text{inv}}^{\phi} \leq 0$, the *Gap Term* is always $\leq 0$. Meanwhile, in the optimization of the gap term, we set $\gamma = 1$ to facilitate sampling and training, so we need to optimize $\phi, \theta$ by maximizing the gap term as follows:

$$\phi^*, \theta^* = \arg\max J_{\text{Gap Term}}(\phi, \theta)$$

$$= \arg\max_{\phi, \theta} \mathop{\mathbb{E}}_{\substack{(s_t, a_t, s_{t+1}) \sim \pi \\ s_{t+1} \sim P(\cdot \mid s_t, a_t)}} \left[ p_{\text{inv}}^{\phi}(e = f^{-1}(a_t; \theta) \mid s_t, s_{t+1}) \right] \tag{10}$$

Specifically, we use the iterative optimization mechanism to optimize $\phi$ and $\theta$ for the given inner policy $\pi_i$. Let $\phi^k$ and $\theta^k$ denote the learned parameters after iteration $k$, then:

**Step 1.** Given $\theta = \theta^k$, we optimize the objective function in Equation 10 by optimizing $\phi$:

$$\phi^{k+1} = \arg\max_\phi J_{\text{Gap Term}}(\phi, \theta^k)$$

$$= \arg\min_\phi \mathop{\mathbb{E}}_{(s,s',e)\in\mathcal{D}} - \log\left[p_{\text{inv}}^\phi(e \mid s, s')\right] \quad (11)$$

where $\mathcal{D}$ denotes the dataset collected by policy $\pi$. Intuitively, minimizing the objective in Equation 11 amounts to maximum likelihood estimation of actions.

**Step 2.** Given $\phi = \phi^{k+1}$, we optimize the objective function in Equation 10 by optimizing $\theta$:

$$\theta^{k+1} = \arg\max_\theta J_{\text{Gap Term}}(\phi^{k+1}, \theta)$$

$$\text{where} \quad a = f(e; \theta), \quad \pi(a \mid s) = \pi_i(e \mid s) * \left|\frac{\partial f(e;\theta)}{\partial e}\right|^{-1}. \quad (12)$$

It is noteworthy that the action mapping layer $f_\theta$ is implicitly included in $\pi$, preventing direct optimization of $\theta$. To solve this issue, we adopt the gradient descent method as follows:

**Theorem 5.2.** *Given the inverse dynamic $p_{\text{inv}}^{\phi^{k+1}}(e \mid s, s')$, the gradient of $J_{\text{Gap Term}}(\theta)$ can be derived as:*

$$\nabla_\theta J_{\text{Gap Term}}(\theta) = \mathop{\mathbb{E}}_{\substack{s_0\in\mathcal{S}, a_t\sim\pi(\cdot|s_t) \\ s_{t+1}\sim P(\cdot|s_t,a_t)}}\left[\nabla_\theta \log\left|\frac{\partial f(e;\theta)}{\partial e}\right|^{-1}\Big|_{e=e_t} \log p_{\text{inv}}^{\phi^{k+1}}(e_t \mid s_t, s_{t+1})\right], \quad (13)$$

*where $\left|\frac{\partial f(e;\theta)}{\partial e}\right|$ is the Jacobian determinant of the function $f$ and $e_t = f^{-1}(a_t; \theta)$.*

*Proof.* Please refer to Appendix D for the detailed proof. $\qquad\square$

Based on Theorem 5.2, we can perform gradient updates on $\theta$ to train the action mapping layer $f_\theta$.

## 5.3 PRACTICAL IMPLEMENTATION

The overall framework of our algorithm is illustrated in Algorithm 1. After interacting with the environment, we iteratively train the inverse dynamics network $p_{\text{inv}}^\phi(e \mid s, s')$ and the action mapping layer $f_\theta(e)$ using the collected data based on the Equation 11 and Theorem 5.2. For the inner policy $\pi_i$, we use the standard maximum policy entropy RL methods, such as SAC (Haarnoja et al., 2018c).

Specifically, we construct the invertible action mapping function $f$ using a piecewise linear function with $N$ parameters ($\vec{\theta} \in \mathbb{R}^N$), defined as follows:

$$f(x) = \sum_{j=1}^{i-1} k_j \cdot \frac{1}{N} + k_i\left(x - \frac{i-1}{N}\right) \quad \text{for } x \in \left[\frac{i-1}{N}, \frac{i}{N}\right]$$

where $k_i = N \cdot \frac{\exp(\theta_i)}{\sum_{j=1}^N \exp(\theta_j)}$ for $i = 1, 2, \ldots, N$. For each $i \in [1, N]$, $k_i$ represents the slope of the linear function in the interval $\left[\frac{i-1}{N}, \frac{i}{N}\right]$. For environments with multidimensional action spaces, we construct $|\mathcal{A}|$ action mapping functions, each applying an independent transformation to its respective dimension.

For the inverse dynamics model, rather than using Gaussian distributions to predict the distribution of continuous actions, we discretize the actions in the dataset and employ discrete multinomial distributions. These multinomial distributions output an $M$-dimensional vector, where each dimension corresponds to the probability that the predicted action lies within the interval $\left[\frac{j-1}{M}, \frac{j}{M}\right]$, for $j \in [1, M]$. This approach enables the inverse dynamics model to capture complex and multimodal behaviors effectively.

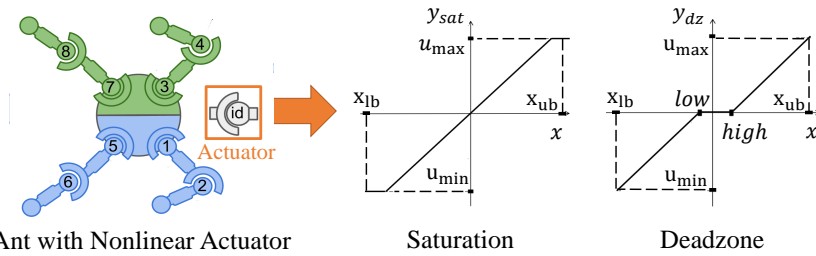

Figure 2: The control of systems with actuators demonstrating input nonlinearities (e.g., saturation, deadzone).

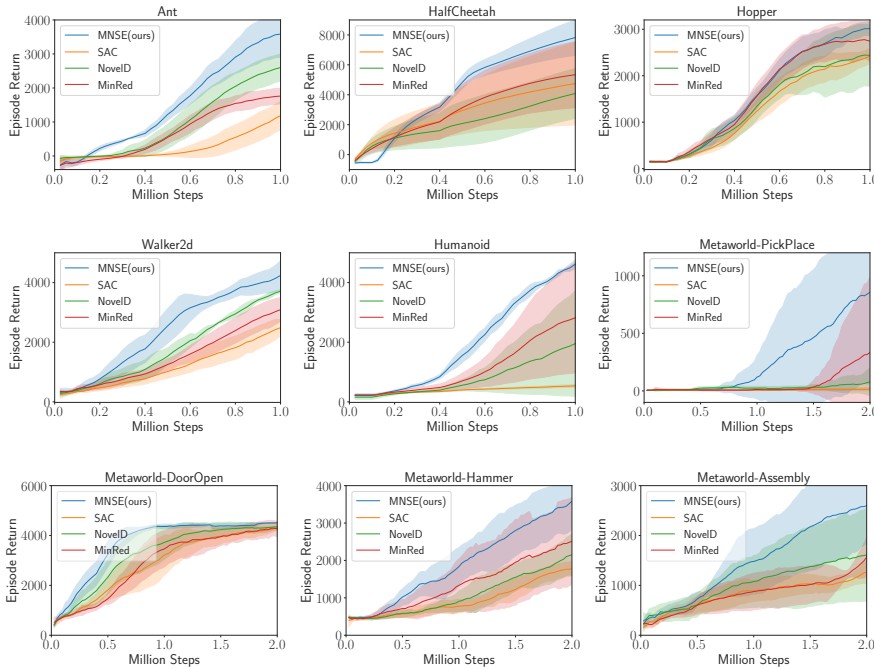

Figure 3: The experimental results in MuJoCo and MetaWorld with nonlinear actuators under five random seeds.

## 6 EXPERIMENTS

In this section, we aim to address the following questions: (1) How do traditional RL methods perform when the policy entropy does not accurately reflect the diversity of the state? (2) How does MNSE compare with other state-of-the-art approaches for maximizing entropy? (3) How sensitive is MNSE to the hyperparameters of the algorithm?

### 6.1 EXPERIMENTAL SETTING

In this section, we examine actuators with input nonlinearities. In industrial applications, nonlinear actuators are a common challenge due to wear-and-tear or inaccuracies in mechanical components. In this work, we consider two types of input nonlinearities: saturation and deadzone, as shown in Figure 2. Both saturation and deadzone can reduce control accuracy, severely influencing the control

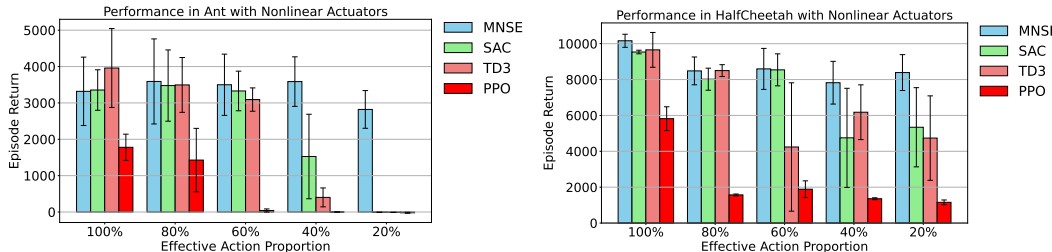

Figure 4: **(Left)**: Perfomance of MNSE, SAC, TD3, PPO in MuJoCo Ant with nonlinear actuators under different effective action proportions. **(Right)**: Perfomance of MNSE, SAC, TD3, PPO in MuJoCo HalfCheetah with nonlinear actuators under different effective action proportions.

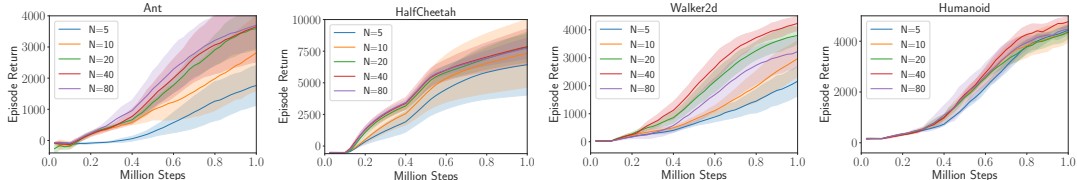

Figure 5: Ablation study of the number of parameters in the piecewise linear function.

system's performance. Specifically, the saturation and deadzone are formulated as follows:

$$
y_{\text{sat}} = \begin{cases} u_{\max} & \text{if } x \geq u_{\max} \\ x & \text{if } u_{\min} < x < u_{\max} \\ u_{\min} & \text{if } u \leq u_{\min} \end{cases} \qquad y_{\text{dz}} = \begin{cases} x - \text{high} & \text{if } u \geq \text{high} \\ 0 & \text{if low } < u < \text{high} \\ x - \text{low} & \text{if } u \leq \text{low} \end{cases}
$$

where $x \in [x_{\text{lb}}, x_{\text{ub}}], y \in [u_{\min}, u_{\max}]$. Notably, within a specific environment, $u_{\min}$ and $u_{\max}$ remain invariant, guaranteeing theoretical optimality across various EAPs.

In the experiments, we employ either saturation or deadzone to each joint of the robot. In addition, we characterize the proportion of effective actions in the action space as Effective Action Proportion (EAP):

$$
\text{EAP} = \frac{u_{\max} - u_{\min}}{x_{\text{ub}} - x_{\text{lb}}}. \tag{14}
$$

As EAP increases, the actuator nonlinearity increases, making the task more difficult.

**Baselines** We compare MNSE with baselines using various maximum entropy methods. We first compare our method with the standard maximum policy entropy method, SAC (Haarnoja et al., 2018c;d). In addition, we compare our method with MinRed (Baram et al., 2021), which directly maximizes the next-state entropy to minimize action redundancy. We also compare our method with the strong state-novelty-based exploration method (Zhang et al., 2021). Please refer to Appendix E for the experimental details.

## 6.2 EXPERIMENTAL RESULTS

**Answer of Question 1:** To show the impact of the nonlinear actuators on the traditional RL methods, we conduct experiments in MuJoCo environments with various EAP by employing SAC (Haarnoja et al., 2018c;d), TD3 (Silver et al., 2014; Lillicrap, 2015), PPO (Schulman et al., 2017), and our proposed MNSE approach. As shown in Figure 4, as the EAP decreases and actuator nonlinearity increases, the performance of SAC gradually degrades and eventually collapses. In the Ant environment (Figure 4, left), when the EAP decreases to 40%, the performance of the SAC algorithm shows a marked decline, and at 20% EAP or lower, SAC completely fails. Similarly, in the HalfCheetah environment (Figure 4, right), when the EAP drops to 40% or lower, the performance of SAC is only half of that achieved without actuator nonlinearity (EAP = 100%).

This decline in performance can be attributed to the fact that as EAP decreases, policy entropy no longer accurately represents the diversity of the next state, which hinders the effective exploration guidance within the SAC framework. In contrast, our MNSE approach based on next-state entropy maintains stable performance across varying EAP environments, significantly outperforming SAC. Moreover, our experiments indicate that not only SAC, but traditional RL algorithms such as TD3 and PPO, also struggle to handle the reduction in effective action space as actuator nonlinearity increases. As shown in Figure 4 (right), when the EAP decreases, both TD3 and PPO exhibit varying degrees of performance degradation, with some cases resulting in complete failure. This underscores that actuator nonlinearity is a significant challenge across various algorithms in the field of reinforcement learning.

**Answer of Question 2:** To show that MNSE can maximize next-state entropy, we conduct experiments on Mujoco and MetaWorld tasks. As illustrated in Figure 3, our method consistently outperforms baseline approaches across various experimental environments. NovelD and MinRed employ bonus-based strategies to promote the exploration of diverse states. Specifically, NovelD incentivizes agents by evaluating state novelty, while MinRed provides additional rewards based on transition entropy. In contrast, our method, MNSE, establishes an action mapping layer that effectively bridges the gap between policy entropy and next-state entropy. By directly promoting exploration through next-state entropy, MNSE demonstrates superior performance compared to bonus-based methods across various domains, as evidenced by our experimental results.

**Answer of Question 3:** To test how the algorithm's hyperparameters affect the performance of MNSE, we change the number of parameters $N$ in the piecewise linear function, which significantly influences the expressive power of the action mapping function $f$. As shown in Figure 5, we conducted ablation experiments across four MuJoCo environments to evaluate the impact of $N$. The results reveal that when $N$ is small, the limited expressive capacity of $f$ leads to suboptimal algorithm performance. As $N$ increases, algorithm performance gradually improves. However, once $N \geq 20$, the performance stabilizes and shows little variation. To balance expressive power with computational efficiency, we consistently employed $N = 20$ in all experimental implementations.

## 7 DISCUSSION

*Why Maximize Next-State Entropy in Reinforcement Learning?* Entropy regularization is a fundamental technique in reinforcement learning. By integrating an entropy maximization term, it enhances robustness to model and estimation errors (Ziebart et al., 2010), promotes the acquisition of diverse behaviors (Haarnoja et al., 2017), facilitates broader exploration (Fox et al., 2015; Haarnoja et al., 2018c;d) and accelerates the learning process by smoothing the optimization landscape (Ahmed et al., 2019). However, maximizing policy entropy may not directly promote policy optimization due to redundancy in the action space. In such cases, next-state entropy extends the concept of policy entropy more directly. Specifically, next-state entropy measures the entropy of the next state resulting from the policy, rather than the action itself. This shift allows next-state entropy to capture the diversity of effects induced by actions. By bridging the gap between next-state and policy entropy, our method retains the benefits of policy entropy while addressing inefficiencies caused by action redundancy.

## 8 CONCLUSION

In this work, we demonstrate a critical problem: the maximum next-state entropy of the RL agent. We first systematically elucidate the distinctions and interrelationships between next-state entropy and policy entropy. Then, to bridge the gap between these two concepts, we integrate inverse dynamics with an action mapping layer. We demonstrate that by optimizing both the action mapping layer and inverse dynamics model, we can maximize the next-state entropy by optimizing the inner policy entropy. We conduct extensive experiments and demonstrate that our method outperforms baseline methods across various domains. Future research will focus on extending MNSE to accommodate more complex action space structures and exploring its potential applications in robotics.

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

## A    TOY EXAMPLE

Consider a one-step MDP with a deterministic transition, where $s_{t+1} = \max(a_t, 0)$, which means actions less than zero are redundant. We assume an initial policy $\pi$ is a Gaussian policy with a mean less than zero: $p(x) = \frac{1}{\sqrt{2\pi}\sigma} \exp\left(-\frac{(x-\mu)^2}{2\sigma^2}\right), \mu < 0$. The corresponding policy entropy is given by:

$$\mathcal{H}(\pi(\cdot \mid s_t)) = \log(\sqrt{2\pi e}\sigma), \tag{15}$$

which is independent of the mean $\mu$.

Since $P^\pi(s' \mid s) = 0$ when $s' < 0$, it follows that $P^\pi(s' \mid s)$ is a rectified Gaussian distribution. Then we have:

$$
\begin{aligned}
&\mathcal{H}(S_{t+1} \mid S_t = s, \pi) \\
&= -\int_0^{+\infty} p(x) \log p(x) dx \\
&= -\int_0^{+\infty} \frac{\varphi(\frac{x-\mu}{\sigma})}{\sigma} \log \frac{\varphi(\frac{x-\mu}{\sigma})}{\sigma} dx \\
&= -\int_{-\frac{\mu}{\sigma}}^{+\infty} \varphi(z) \log \frac{\varphi(z)}{\sigma} dz \\
&= \int_{-\frac{\mu}{\sigma}}^{+\infty} -\varphi(z) \log \varphi(z) dz + \log \sigma \int_{-\frac{\mu}{\sigma}}^{+\infty} \varphi(z) dz \\
&= \int_{-\infty}^{+\infty} -\varphi(z) \log \varphi(z) dz - \int_{-\infty}^{-\frac{\mu}{\sigma}} -\varphi(z) \log \varphi(z) dz + \log \sigma (1 - \Phi(-\frac{\mu}{\sigma})) \\
&= \underbrace{\log(\sqrt{2\pi e}\sigma)}_{\text{policy entropy}} + \left\{ \Phi(\frac{\mu}{\sigma}) \log \sigma - \int_{-\infty}^{-\frac{\mu}{\sigma}} -\varphi(z) \log \varphi(z) dz \right\},
\end{aligned}
\tag{16}
$$

where $\varphi(z) = \frac{e^{-z^2/2}}{\sqrt{2\pi}}$ is the probability density function of the standard normal distribution, and $\Phi(x) = \int_{-\infty}^{x} \varphi(z) dz$ is its cumulative distribution function.

It is challenging to provide a rigorous mathematical analysis of this relationship. Specifically, when we assume $\sigma > 1$ in the toy example, $\log \sigma > 0$, and $\Phi\left(\frac{\mu}{\sigma}\right) \log \sigma$ is positively correlated with $\mu$. Furthermore, since $0 < \varphi(z) \le \frac{1}{\sqrt{2\pi}} < 1$, the term $-\int_{-\infty}^{-\frac{\mu}{\sigma}} -\varphi(z) \log \varphi(z) \, dz$ is also positively correlated with $\mu$. Meanwhile, we have conducted numerical experiments to examine the relationship between the $\mu$ variable and next-state entropy for different $\sigma$ values. As shown in the results, when $\sigma$ values are $\{1.0, 1.5, 2.0\}$, the next-state entropy increases as the $\mu$ value increases, demonstrating a positive relationship between $\mu$ and entropy.

The above analysis indicates that to increase $\mathcal{H}(S_{t+1} \mid S_t = s, \pi)$, one effective method is not only to increase $\sigma$ but also to increase $\mu$.

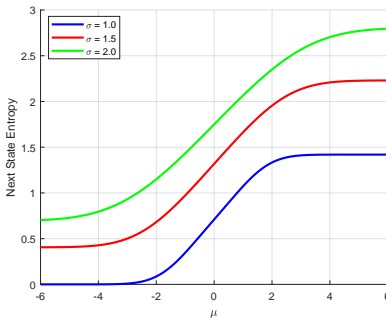

Figure 6: The next-state entropy increases as the mu value increases.

## B   PROOF OF THEOREM  4.4

*Proof.* Recall that the entropy of state is:

$$\mathcal{H}(S_{t+1} \mid S_t = s, \pi) = -\mathbb{E}_{a \sim \pi(\cdot|s)}\mathbb{E}_{s' \sim P(\cdot|s,a)} \log\left[P^\pi(s' \mid s)\right]$$
$$= -\mathbb{E}_{a \sim \pi(\cdot|s)}\mathbb{E}_{s' \sim P(\cdot|s,a)} \log\left[\mathbb{E}_{a \sim \pi(\cdot|s)}P(s' \mid s,a)\right] \qquad (17)$$

Then, according to the Definition 4.3, we have:

$$\mathbb{E}_{s' \sim P^\pi(\cdot|s,a)} \log\left[\mathbb{E}_{a \sim \pi(\cdot|s)}P(s' \mid s,a)\right] = \mathbb{E}_{s' \sim P^\pi(\cdot|s,a)} \log\left[\pi(a \mid s)P(s' \mid s,a)\right]$$
$$= \mathbb{E}_{s' \sim P^\pi(\cdot|s,a)}\left[\log \pi(a \mid s) + \log P(s' \mid s,a)\right], \qquad (18)$$

Combining the Equation 17 and Equation 18, we have:

$$\mathcal{H}(S_{t+1} \mid S_t = s, \pi) = -\mathbb{E}_{a \sim \pi(\cdot|s)}\mathbb{E}_{s' \sim P^\pi(\cdot|s,a)}\left[\log \pi(a \mid s) + \log P(s' \mid s,a)\right]$$
$$= -\mathbb{E}_{a \sim \pi(\cdot|s)}\left[\log \pi(a \mid s)\right] - \mathbb{E}_{a \sim \pi(\cdot|s)}\mathbb{E}_{s' \sim P^\pi(\cdot|s,a)}\left[\log P(s' \mid s,a)\right] \quad (19)$$
$$= \mathcal{H}(\pi(\cdot \mid s)) + \mathbb{E}_{a \sim \pi(\cdot|s)}\mathcal{H}(S_{t+1} \mid S_t = s,a).$$

$\square$

## C   PROOF OF THEOREM  5.1

*Proof.* Firstly, we can derive the next-state entropy $\mathcal{H}\left(S_{t+1} \mid S_t = s, \pi\right)$ as follows.

$$\mathcal{H}\left(S_{t+1} \mid S_t = s, \pi\right) = -\mathbb{E}_{a \sim \pi(\cdot|s)}\mathbb{E}_{s' \sim p(\cdot|s,a)} \log p^\pi(s' \mid s)$$
$$= -\mathbb{E}_{s' \sim p^{\pi_i}(\cdot|s)}\mathbb{E}_{e \sim p_{\text{inv}}(\cdot|s,s')} \log p^{\pi_i}(s'|s)$$
$$= -\mathbb{E}_{s' \sim p^{\pi_i}(\cdot|s)}\mathbb{E}_{e \sim p_{\text{inv}}(\cdot|s,s')} \left\{\log p^{\pi_i}(s'|s) - \log\left[\pi_i(e|s)p(s'|s,e)\right] + \log\left[\pi_i(e|s)p(s'|s,e)\right]\right\}$$
$$(20)$$

where

$$p^\pi(s' \mid s) = \mathbb{E}_{\tilde{a} \sim \pi(\cdot|s)}p\left(s' \mid s, \tilde{a}\right), \quad p^{\pi_i}(s' \mid s) = \mathbb{E}_{\tilde{e} \sim \pi_i(\cdot|s)}p\left(s' \mid s, \tilde{e}\right).$$

Here, we utilize the property of inverse dynamics:

$$p_{\text{inv}}(e|s, s') = \frac{\pi_i(e|s)p(s'|s,e)}{p^{\pi_i}(s'|s)} \qquad (21)$$

then Eq. 20 can be derived as follows:

$$\mathcal{H}\left(S_{t+1} \mid S_t = s, \pi\right)$$
$$= -\mathbb{E}_{s' \sim p^{\pi_i}(\cdot|s)}\mathbb{E}_{e \sim p_{\text{inv}}(\cdot|s,s')} \left\{\log\left[\pi_i(e|s)p(s'|s,e)\right] + \log p^{\pi_i}(s'|s) - \log\left[\pi_i(e|s)p(s'|s,e)\right]\right\}$$
$$= -\mathbb{E}_{s' \sim p^{\pi_i}(\cdot|s)}\mathbb{E}_{e \sim p_{\text{inv}}(\cdot|s,s')} \log\left[\pi_i(e|s)p(s'|s,e)\right] -$$
$$\qquad \mathbb{E}_{s' \sim p^{\pi_i}(\cdot|s)}\mathbb{E}_{e \sim p_{\text{inv}}(\cdot|s,s')} \left\{\log p^{\pi_i}(s'|s) - \log\left[\pi_i(e|s)p(s'|s,e)\right]\right\} \quad \text{(Property of Eq. 21)}$$
$$= -\mathbb{E}_{s' \sim p^{\pi_i}(\cdot|s)}\mathbb{E}_{e \sim p_{\text{inv}}(\cdot|s,s')} \log\left[\pi_i(e|s)p(s'|s,e)\right] - \mathbb{E}_{s' \sim p^{\pi_i}(\cdot|s)}\mathbb{E}_{e \sim p_{\text{inv}}(\cdot|s,s')} \left\{-\log p_{\text{inv}}(e|s, s')\right\}$$
$$= -\mathbb{E}_{s' \sim p^{\pi_i}(\cdot|s)}\mathbb{E}_{e \sim p_{\text{inv}}(\cdot|s,s')} \log\left[\pi_i(e|s)p(s'|s,e)\right] - \mathbb{E}_{e \sim \pi_i(\cdot|s)}\mathbb{E}_{s' \sim p(\cdot|s,e)} \left\{-\log p_{\text{inv}}(e|s, s')\right\}$$
$$= -\mathbb{E}_{s' \sim p^{\pi_i}(\cdot|s)}\mathbb{E}_{e \sim p_{\text{inv}}(\cdot|s,s')} \log\left[\pi_i(e|s)p(s'|s,e)\right] + \mathbb{E}_{a \sim \pi(\cdot|s)}\mathbb{E}_{s' \sim p(\cdot|s,a)} \left[\log\left[p_{\text{inv}}(e \mid s, s')\right]\right].$$

The first term is as follows:

$$-\mathbb{E}_{s' \sim p^{\pi_i}(\cdot|s)}\mathbb{E}_{e \sim p_{\text{inv}}(\cdot|s,s')} \log\left[\pi_i(e|s)p(s'|s,e)\right]$$
$$= -\mathbb{E}_{e \sim \pi_i(\cdot|s)}\mathbb{E}_{s' \sim p(\cdot|s,e)} \log\left[\pi_i(e|s)p(s'|s,e)\right]$$
$$= -\mathbb{E}_{e \sim \pi_i(\cdot|s)}\mathbb{E}_{s' \sim p(\cdot|s,e)} \log \pi_i(e|s) - \mathbb{E}_{e \sim \pi_i(\cdot|s)}\mathbb{E}_{s' \sim p(\cdot|s,e)} \log p(s'|s,e)$$
$$= \mathbb{E}_{e \sim \pi_i(\cdot|s)}\left[-\log \pi_i(e|s)\right] + \mathbb{E}_{e \sim \pi_i(\cdot|s)}\mathbb{E}_{s' \sim p(\cdot|s,e)}\left[-\log p(s'|s,e)\right]$$
$$= \mathcal{H}\left(\pi_i(\cdot \mid s)\right) + \mathcal{H}_{\text{model}}$$

where $\mathcal{H}_{\text{model}} = \mathbb{E}_{e \sim \pi_i(\cdot|s)} [\mathcal{H}(S_{t+1} \mid S_t = s, e)]$ is the entropy of the dynamics model.

As a result, we have:

$$\mathcal{H}(S_{t+1} \mid S_t = s, \pi) = \mathcal{H}(\pi_i(\cdot \mid s)) + \underbrace{\mathbb{E}_{a \sim \pi(\cdot|s)} \mathbb{E}_{s' \sim p(\cdot|s,a)} [\log [p_{\text{inv}}(e \mid s, s')]]}_{\text{Gap Term}} + \mathcal{H}_{\text{model}}$$

$\square$

## D    PROOF OF THEOREM 5.2

*Proof.* We will derive $\nabla_\theta J_{\text{Gap Term}}(\theta)$ in the following.

$$\nabla_\theta \mathop{\mathbb{E}}_{\substack{s_0 \in \mathcal{S}, a_t \sim \pi(\cdot|s_t) \\ s_{t+1} \sim P(\cdot|s_t, a_t)}} \left[ p_{\text{inv}}^\phi(e = f^{-1}(a_t; \theta) \mid s_t, s_{t+1}) \right] = \int_{s \in \mathcal{S}} \int_{a \in \mathcal{A}} \frac{\partial \pi(a|s)}{\partial \theta} \mathbb{E}_{s' \sim p(\cdot|s,a)} \log p_{\text{inv}}^{\phi^{k+1}}(e \mid s, s')$$

$$= \int_{s \in \mathcal{S}} \int_{a \in \mathcal{A}} \pi(a|s) \frac{1}{\pi(a|s)} \frac{\partial \pi(a|s)}{\partial \theta} \mathbb{E}_{s' \sim p(\cdot|s,a)} \log p_{\text{inv}}^{\phi^{k+1}}(e \mid s, s')$$

$$= \int_{s \in \mathcal{S}} \int_{a \in \mathcal{A}} \pi(a|s) \frac{\partial \log \pi(a|s)}{\partial \theta} \mathbb{E}_{s' \sim p(\cdot|s,a)} \log p_{\text{inv}}^{\phi^{k+1}}(e \mid s, s')$$

$$= \mathop{\mathbb{E}}_{\substack{s_0 \in \mathcal{S}, a_t \sim \pi(\cdot|s_t) \\ s_{t+1} \sim P(\cdot|s_t, a_t)}} \frac{\partial \log \pi(a|s)}{\partial \theta} \log p_{\text{inv}}^{\phi^{k+1}}(e \mid s, s')$$

when $a = f_\theta(e)$, recall that:

$$\pi(a|s) = \pi_i(e \mid s) * \left| \frac{\partial a}{\partial e} \right|^{-1},$$

According to the inverse function theorem: If $y = f(x)$ and $x = f^{-1}(y)$, we have:

$$\frac{df^{-1}(y)}{dy} = \frac{dx}{dy} = \left( \frac{dy}{dx} \right)^{-1} = \left( \frac{df(x)}{dx} \right)^{-1}$$

then:

$$\frac{\partial \log \pi(a|s)}{\partial \theta} = \frac{\partial \log |\frac{\partial f_\theta^{-1}(a)}{\partial a}|}{\partial \theta} = \nabla_\theta \log \left| \frac{\partial f_\theta(e)}{\partial e} \right|^{-1}$$

In conclusion,

$$\nabla_\theta J_{\text{Gap Term}}(\theta) = \mathop{\mathbb{E}}_{\substack{s_0 \in \mathcal{S}, a_t \sim \pi(\cdot|s_t) \\ s_{t+1} \sim P(\cdot|s_t, a_t)}} \left\{ \nabla_\theta \log \left| \frac{\partial f(e; \theta)}{\partial e} \right|^{-1} \bigg|_{e = e_t} \log p_{\text{inv}}^{\phi^{k+1}}(e_t \mid s_t, s_{t+1}) \right\} \tag{22}$$

$\square$

# E  EXPERIMENTAL DETAILS

Our algorithm, MNSE, is developed based on the SAC algorithm from the RL Baselines3 Zoo (Raffin, 2020; Raffin et al., 2021). The hyperparameters for MNSE are detailed in Table E. In all experimental implementations, we consistently employed $N = 20$ as the number of parameters in the piecewise linear function. The baseline SAC shares the same hyperparameters as those of MNSE.

| Hyper-parameter | Value |
|---|---|
| **Shared** | |
| Learning rate | $3 \times 10^{-4}$ |
| Buffer size | $1 \times 10^{6}$ |
| Learning starts | $1 \times 10^{5}$ |
| Batch size | 64 |
| Soft update coefficient $\tau$ | 0.005 |
| Discount factor $\gamma$ | 0.99 |
| Activation function | ReLU |
| **Others** | |
| Number of parameters in piecewise linear function $N$ | 20 |

Table 1: MNSE Hyper-parameters sheet

**Baseline Hyper-parameter:** TD3 and PPO are developed utilizing the RL Baselines3 Zoo (Raffin, 2020), employing the tuned hyperparameters provided by this framework. NovelD (Zhang et al., 2021) and MinRed (Baram et al., 2021) are constructed based on the SAC algorithm within the RL Baselines3 Zoo, with the trade-off coefficient for the additional reward being searched within the range of 5e-3 to 5e-1.

**Inverse Dynamics:** We use discrete multinomial distributions to predict the actions. These multinomial distributions output an $M$-dimensional vector, where each dimension corresponds to the probability that the predicted action lies within the interval $\left[\frac{j-1}{M}, \frac{j}{M}\right]$, for $j \in [1, M]$. We set $M = 20$ in all experimental implementations.

**Nonlinear Actuator in Figure 3:** We consider two types of input nonlinearities: saturation and deadzone. The specific types of input nonlinearities and the corresponding effective action proportions (EAP) in each environment during our experiments(Figure 3) are presented in Table E.

| Environment | Types of Nonlinearities | EAP |
|---|---|---|
| MuJoCo Ant | Saturation | 40% |
| MuJoCo HalfCheetah | Saturation | 40% |
| MuJoCo Hopper | Deadzone | 20% |
| MuJoCo Walker2d | Deadzone | 20% |
| MuJoCo Humanoid | Deadzone | 20% |
| Metaworld PickPlace | Saturation | 40% |
| Metaworld Hammer | Saturation | 40% |
| Metaworld DoorOpen | Deadzone | 40% |
| Metaworld Assembly | Deadzone | 40% |

Table 2: Nonlinear Actuator in Figure 3

