# OpenReview forum: "Maximum Next-State Entropy for Efficient Reinforcement Learning"
_ICLR.cc/2025/Conference — ICLR 2025 Conference Withdrawn Submission_

### Official Review · Reviewer_ybLa · 2024-10-28

**Soundness:** 3
**Presentation:** 3
**Contribution:** 2
**Rating:** 6
**Confidence:** 5

**Summary:**

This paper theoretically highlights the distinction between policy entropy and next-state entropy in Markov Decision Processes (MDPs) and makes a compelling argument that the two entropies are equivalent if the policy is non-redundant---meaning that different actions lead to different next states given the same state in the MDP. The paper then shifts its focus to demonstrating the advantages of incorporating maximum next-state entropy into the reinforcement learning process in MDPs. This is done by deliberately introducing saturation and deadzone effects into the control system to create redundant policies. Numerous experiments demonstrate their method can outperform the baselines.

**Strengths:**

The paper is well-written and clear, with a solid theoretical analysis and a sufficient set of experiments.

**Weaknesses:**

1. In the introduction, the authors present equipment aging as an example of redundancy in the action space. However, this example does not fully convince the reviewer. Specifically, reinforcement learning assumes that the Markov Decision Process (MDP) remains consistent. When changes occur in the action space, such as those caused by aging equipment, the previously learned policy may no longer perform effectively within the altered MDP. Further clarification or a more suitable example might strengthen this argument.

2. There is an abuse of notation for reward function r(s,a) while the paper assumes the reward is only affected by states.

**Questions:**

1. Based on the theory, when effective action proportion is 100%, the MNSE should have the same performance as SAC (the base model adopted by MNSE). But in Fig 4. there are some differences, any explanation or analysis on this observation?

2. For eq (4), how can we guarantee the mu variable is positive related with the entropy? The reviewer did not see any further analysis on this part (not even in the appendix).

3. All experiments are conducted in the continuous action space, will maximum next-state entropy benefit the policy learning in discrete action space environments?

4. Can you explain in more details on how to derive the content of equation (8)?

5. To minimize the gap term in Theorem 5.1, step 1 and 2 are provided in equation (11) and (12). Why the parameters of the inverse dynamic of inner policy are optimized first, instead of the ones of the mapping layer?

---

> ### Author Response · Authors · 2024-11-22
> **Response to Reviewer ybLa**
>
> Dear Reviewer,
>
> Thanks for your valuable comments.
> We hope the following statement can address your concern.
>
> **W1: Further clarification of example on aging equipment in introduction.**
>
> **A for W1:**
> Yes, we have revised this section in the introduction to provide further clarification.
> In our setup, before learning the policy, the equipment already has saturation or deadzone effects (caused by aging equipment or design redundancy). During policy learning and deployment, the action space remains unchanged.
>
> Our work focuses on policy learning when there is redundancy in the action space, rather than on adapting the policy when the action space itself changes.
>
>
> **W2:  Abuse of notation for reward function.**
>
> **A for W2:**
> Yes, we have modified the notation to ensure consistency throughout the paper.
>
>
> **Q1: There are some differences between MNSE and SAC when effective action proportion is 100\%**
>
> **A for Q1:**
> Yes, there are differences in performance between MNSE and SAC when the effective action proportion (EAP) is 100\%, and they are not exactly equal. There are two reasons for this:
>
> * Even when the actuator's torque and the input commands are perfectly linear, the change in the state is not necessarily linear. MNSE is designed to eliminate such redundancy, leading to better performance in tasks like HalfCheetah.
> * At EAP=100\%, due to model learning and gradient training, the action mapping layer learned by MNSE may have a slight discrepancy from the true identity function (where actions = inner actions), which results in small performance differences, as observed in tasks like Ant.
>
> **Q2:  How can we guarantee the mu variable is positive related with the entropy?**
>
> **A for Q2:**
> Providing a rigorous mathematical analysis is challenging.
> However, we have included numerical results for different sigma values, as shown in Figure. 6 in the Appendix. A.
> The results indicate that next-state entropy increases with higher mu values, demonstrating a positive correlation between mu and entropy.
>
> **Q3:  Will maximum next-state entropy benefit the policy learning in discrete action space environments?**
>
> **A for Q3:**
> As discussed in Section 4,
> in discrete action spaces, the fewer redundant actions there are, the smaller the gap between next-state entropy and policy entropy.
> This insight suggests that evaluating actions and removing those with identical effects within the discrete action set can help improve exploration efficiency.
>
> **Q4: Can you explain in more details on how to derive the content of equation (8)**
>
> **A for Q4:**
> The content of equation (8) is derived using the \textit{Change of Variable Theorem}.
>
> The Change of Variable Theorem states that if you have a random variable $ X $ with a probability density function $p_X(x)$, and you apply a transformation $ y = g(x) $, then the probability density of the new variable $ Y $ can be obtained by adjusting the original density using the Jacobian of the transformation:
>
> $$
> p\_Y(y) = p\_X(g^{-1}(y)) \left| \frac{d}{dy} g^{-1}(y) \right|
> $$
>
>
> **Q5: Why the parameters of the inverse dynamic of inner policy are optimized first, instead of the ones of the mapping layer?**
>
> **A for Q5:**
> This is a technical decision related to the initialization process. At the beginning, the inverse dynamics network is randomly initialized, while the mapping layer is initialized as an identity function (i.e., actions = inner actions). Optimizing the inverse dynamics network first helps avoid the negative impact of the random initialization of the inverse dynamics network on the updates to the mapping layer. By optimizing the inverse dynamics network first, we ensure that the mapping layer can later benefit from a more stable inverse dynamics model, leading to more effective and efficient learning.

---

> > ### Comment · Reviewer_ybLa · 2024-11-24
> >
> > Thank you for the authors' response! Considering the potential challenges in providing a rigorous mathematical analysis for Q2, the reviewer suggests revising the corresponding section to incorporate a heuristic assumption as the basis for the subsequent analysis

---

> ### Author Response · Authors · 2024-11-25
> **Response to Reviewer ybLa**
>
> Dear Reviewer,
>
> Thank you for the reviewer’s valuable suggestion! We have added the assumption $\sigma>1$ in the toy example (Section 4.1, Line 184). Based on this assumption, we have provided theoretical analysis and numerical results in Appendix A (Line 734) to demonstrate that the
> $\mu$ variable is positively related to the entropy.
>
> We sincerely appreciate your feedback, which has helped make our paper more rigorous.
>
> Best,
>
> The Authors

---

> ### Comment · Reviewer_ybLa · 2024-11-25
>
> Thank you for your efforts! I’ve raised my score accordingly.

---

> > ### Author Response · Authors · 2024-11-25
> > **Thanks for raising the score to 6!**
> >
> > We would like to thank the reviewer for raising the score to 6! We also appreciate the valuable comments, which helped us significantly improve the paper's strengths.

---

### Official Review · Reviewer_pNrB · 2024-11-01

**Soundness:** 3
**Presentation:** 3
**Contribution:** 2
**Rating:** 5
**Confidence:** 3

**Summary:**

This paper presents a thorough theoretical analysis of the relationship between maximizing next-state entropy and policy entropy. The authors propose a novel framework that links these two types of entropy through an innovative approach that utilizes an inner policy and an action mapping function. Based on this theoretical foundation, the authors introduce the Next-State Entropy Maximization algorithm (MNSE), which is shown to be particularly effective in environments with redundant action spaces. This work contributes valuable insights into entropy maximization, bridging next-state novelty concepts with policy design in reinforcement learning.

**Strengths:**

1. The paper offers a fresh perspective on state novelty by theoretically linking next-state entropy and policy entropy. While state novelty algorithms are known to enhance agent performance in various environments, the theoretical analysis of state entropy remains underexplored. This paper addresses this gap by establishing a detailed connection between next-state entropy and policy entropy, achieved through an internal policy and an action mapping function.

2. The authors provide a rigorous and structured proof process, making it easy for readers to follow the logical progression and understand the interplay between the entropies. This systematic approach gives a solid foundation for the proposed MNSE framework, and the clarity of the theoretical contributions makes the complex subject matter more approachable for readers.

3. The MNSE algorithm is an impressive practical outcome of this research, showcasing strong empirical results in environments with redundant action spaces. This suggests that the algorithm could be beneficial for a wide variety of applications where action redundancy exists, offering new avenues for exploration in reinforcement learning.

**Weaknesses:**

1. The paper includes performance comparisons at 20% and 40% EAP, as seen in Experiment 2. However, expanding these comparisons to include higher EAP levels, such as 80%, 60%, and 100%, would be beneficial. Analyzing performance across a broader range of EAP settings could offer a more comprehensive view of the algorithm’s robustness and adaptability to different entropy thresholds.

2. The current experiments effectively demonstrate MNSE’s performance in Mujoco and Meta-World environments. However, adding further experiments focused on pure exploration tasks—such as those found in maze environments or other exploration-heavy scenarios—would be valuable. Such experiments could provide deeper insights into how MNSE's maximization of next-state entropy impacts exploration behavior, highlighting its effectiveness in environments where exploration quality is critical.

3. While MNSE is compared with well-established state novelty and exploration algorithms, such as MinRed and NovelID (both published in 2021), comparisons with more recent approaches (from 2022 or 2023) could further strengthen the relevance and appeal of this work. Including newer algorithms in the comparative analysis could provide a more current context for MNSE’s performance and underscore its competitiveness among recent advancements in state novelty and exploration research.

**Questions:**

1. In the Related Works section, the authors reference several algorithms (such as SAC, DSPG, CBSQL, and max-min entropy frameworks). Could the authors elaborate on why these algorithms were not included in the experiment section? Understanding the selection criteria for comparison could provide further clarity on the position of MNSE within the broader landscape of state and policy entropy methods.

2. SAC is used as the baseline for updating the policy entropy term in the MNSE framework. It would be interesting to learn how MNSE might perform if alternative algorithms, such as DSPG or CBSQL, were used instead. Could the authors discuss potential outcomes or the theoretical basis for choosing SAC over these other algorithms? Insights into how the baseline choice affects MNSE’s performance would be helpful for researchers considering alternative implementations of the framework.

3. The entropy of all visited states has long been an interesting topic, though challenges remain in addressing it within a solid theoretical framework. Could the authors discuss the correlation between next-state entropy and the total entropy of visited states? This could provide further insight into MNSE’s implications for overall state entropy in agent behavior.

---

> ### Author Response · Authors · 2024-11-22
> **Response to Reviewer pNrB**
>
> Dear Reviewer,
>
> Thanks for your valuable comments.
> We hope the following statement can address your concern.
> We add supplementary experimental results as part of our response.
>
> **W1: Analyzing performance across a broader range of EAP settings.**
>
> **A for W1:**
> We conducted additional experiments to analyze the performance at EAP values of 20\%, 40\%, 60\%, 80\%, and 100\%. This analysis provides a more comprehensive view of the algorithm's robustness and adaptability to different entropy thresholds.
> The results is shown as follows:
>
> | **Environment / EAP** | **MNSE (ours)**         | **SAC**               | **NovelD**              | **MinRed**             |
> |------------------------|-------------------------|-----------------------|-------------------------|------------------------|
> | **Ant**               |                         |                       |                         |                        |
> | 100\%                  | 3318.60 ± 940.35       | 3354.58 ± 556.52      | **4517.24 ± 941.31**    | 3076.22 ± 837.44       |
> | 80\%                   | 3590.74 ± 1169.71      | 3477.07 ± 979.73      | **3652.62 ± 307.70**    | 3053.01 ± 852.90       |
> | 60\%                   | **3499.42 ± 842.01**   | 3328.57 ± 544.72      | 3129.77 ± 258.92        | 2639.14 ± 199.72       |
> | 40\%                   | **3586.25 ± 681.38**   | 1526.37 ± 1162.03     | 2600.26 ± 395.85        | 1761.25 ± 229.65       |
> | 20\%                   | **2820.91 ± 518.60**   | -7.60 ± 1.88          | 1219.92 ± 45.86         | 920.87 ± 476.22        |
> | **HalfCheetah**       |                         |                       |                         |                        |
> | 100\%                  | **10162.46 ± 364.30**  | 9535.45 ± 100.47      | 9860.49 ± 255.37        | 8626.04 ± 963.10       |
> | 80\%                   | **8482.84 ± 775.07**   | 8019.85 ± 614.34      | 7110.31 ± 412.67        | 8451.85 ± 877.56       |
> | 60\%                   | **8594.27 ± 1142.66**  | 8540.62 ± 891.91      | 6679.57 ± 818.11        | 6997.12 ± 1844.33      |
> | 40\%                   | **7824.08 ± 1190.98**  | 4750.61 ± 2762.26     | 4082.68 ± 1668.43       | 5340.42 ± 2207.86      |
> | 20\%                   | **8389.90 ± 999.66**   | 5340.42 ± 2207.86     | 4291.25 ± 771.95        | 4442.10 ± 1719.52      |
> | **Hopper**            |                         |                       |                         |                        |
> | 100\%                  | 3265.50 ± 165.32       | 3173.27 ± 207.26      | **3834.36 ± 346.68**    | 2917.46 ± 426.55       |
> | 80\%                   | 3254.78 ± 172.49       | 2750.05 ± 224.12      | **3858.44 ± 436.45**    | 3064.54 ± 794.75       |
> | 60\%                   | **3120.39 ± 158.73**   | 2417.63 ± 457.04      | 2809.51 ± 265.54        | 2753.56 ± 636.98       |
> | 40\%                   | **3082.22 ± 180.65**   | 2394.08 ± 160.21      | 2648.36 ± 199.94        | 2763.01 ± 655.70       |
> | 20\%                   | **3013.21 ± 149.09**   | 2421.67 ± 160.92      | 2439.52 ± 665.22        | 2743.68 ± 434.33       |
>
> Table 1. Performance across a broader range of EAP settings.
>
>
> **W2: Further experiments focused on pure exploration tasks—such as in maze environments would be valuable.**
>
> **A for W2:**
> We conducted additional experiments with EAP=20\% on pure exploration tasks in maze environments, including maze2d-umaze-v0, maze2d-medium-v0, and maze2d-large-v0.
> The results demonstrate that our method outperforms the baseline across these environments.
>
> | environment   | MNSE         | SAC          | NovelD | MinRed |
> |--------|--------------------|--------------------|--------------------|--------------------|
> | maze2d-umaze-v0    | **41.00±1.14** | 35.94±1.34 | 37.32±1.12 |32.66±2.54 |
> | maze2d-medium-v0    | **35.72±1.84** | 26.24±1.00 | 32.60±0.93 |18.14±1.33 |
> | maze2d-large-v0    | **66.92±8.39** | 52.69±2.89 | 59.77±3.28 |53.82±6.87 |
>
> Table 2. Additional experimental results in maze environments.
>
>
> **W3: Comparisons with more recent approaches (from 2022 or 2023) could further strengthen the relevance and appeal of this work.**
>
> **A for W3:**
> Yes, we have added E3B (2022) [1] as a comparison method in our experiments. The results of these comparisons are provided below.
>
> |Method| Ant (EAP=0.4) | HalfCheetah (EAP=0.4) | Hopper (EAP=0.2) | Walker2d (EAP=0.2) | Humanoid (EAP=0.2) |
> |--------------------|--------------------|--------------------|--------------------|--------------------|--------------------|
> | E3B | 2850.22±363.84 | 3862.70±894.38 | 2708.26±683.06 | 3465.53±79.49 | 3174.87±233.20 |
> | MNSE(Ours) | **3586.25±681.38** | **7824.08±1190.98** | **3013.21±149.09** | **4240.30±484.45** | **5431.75±460.25** |
>
> Table 3. Comparisons with E3B(2022) in MuJoCo.

---

> ### Author Response · Authors · 2024-11-22
> **Response to Reviewer pNrB (2)**
>
> **Q1 \& Q2:  Why choose SAC as backbone framework? Why DSPG, CBSQL, and max-min entropy framework were not included as baseline in the experiment section?**
>
> **A for Q1 \& Q2:**
> SAC is a successor to Soft Q-Learning (SQL) and integrates techniques such as double Q-learning and temperature adjusting (based on stable-baselines3). It is one of the most widely used maxEnt RL algorithms in the community. We believe that making improvements and developments based on the SAC algorithm is highly significant for the community.
>
> The DSPG method (2019) introduced pioneering techniques like DoubleSampling into soft Q-learning. However, the SAC framework (based on stable-baselines3, 2021) has already integrated and refined techniques like double Q-learning.
>
> The CBSQL method (2021) introduces pseudo-count density estimation to Soft Q-Learning, enabling more precise temperature adjustment. However, this method was only evaluated in discrete action spaces (Atari), and it does not show significant advantages in complex tasks in continuous action spaces.
>
> The max-min entropy framework (2021) differs from traditional MaxEnt methods by constructing an exploration policy and separating exploration and exploitation. This added complexity makes it difficult to fairly compare this method with ours.
>
>
> **Q3: The correlation between next-state entropy and the total entropy of visited states.**
>
> **A for Q3:**
> Here are the **key differences** between our method (maximizing next-state entropy) and the state entropy maximizing methods mentioned:
>
> The works mentioned in [2, 3, 4, 5, 6] directly construct policies (which can even be deterministic) to achieve better state coverage.
> However, they do not consider the entropy of policies.
>
> In contrast, our approach builds on the entropy of stochastic policies, which not only accelerates learning and prevents premature convergence to suboptimal solutions but also induces a smoother objective that connects solutions and enable the use of larger learning rates [7].
>
> Next-state entropy is an extension of policy entropy in environments with redundant action spaces.
> Policy entropy encourages action diversity while next-state entropy extends this idea to measure the diversity of effects caused by actions.
> Our method inherits the benefits of policy entropy while overcoming inefficiencies caused by action redundancy.
>
> [1] Henaff M, Raileanu R, Jiang M, et al. Exploration via elliptical episodic bonuses[J]. Advances in Neural Information Processing Systems, 2022, 35: 37631-37646.
>
> [2] Lee, L., Eysenbach, B., Parisotto, E., Xing, E., Levine, S., & Salakhutdinov, R. (2019). Efficient exploration via state marginal matching. arXiv preprint arXiv:1906.05274.
>
> [3] Guo, Z. D., Azar, M. G., Saade, A., Thakoor, S., Piot, B., Pires, B. A., ... & Munos, R. (2021). Geometric entropic exploration. arXiv preprint arXiv:2101.02055.
>
> [4] Islam, R., Ahmed, Z., & Precup, D. (2019). Marginalized state distribution entropy regularization in policy optimization. arXiv preprint arXiv:1912.05128.
>
> [5] Hazan, E., Kakade, S., Singh, K., & Van Soest, A. (2019, May). Provably efficient maximum entropy exploration. In International Conference on Machine Learning (pp. 2681-2691). PMLR.
>
> [6] Liu, H., & Abbeel, P. (2021). Behavior from the void: Unsupervised active pre-training. Advances in Neural Information Processing Systems, 34, 18459-18473.
>
> [7] Ahmed Z, Le Roux N, Norouzi M, et al. Understanding the impact of entropy on policy optimization[C]//International conference on machine learning. PMLR, 2019: 151-160.

---

> ### Author Response · Authors · 2024-11-25
> **Looking forward to further comments!**
>
> Dear Reviewer,
>
> We have added additional explanations and experiments for our methods. We are wondering if our response and revision have cleared your concerns. We would appreciate it if you could kindly let us know whether you have any other questions. We are looking forward to comments that can further improve our current manuscript. Thanks!
>
> Best regards,
>
> The Authors

---

> ### Author Response · Authors · 2024-12-02
>
> Dear Reviewer,
>
> Thank you for your thoughtful feedback on our paper. With only two days remaining in the discussion period, we kindly ask that you review our responses to ensure we have fully addressed your concerns. If you find our responses satisfactory, we would greatly appreciate it if you could reconsider your rating/scoring.
>
> Your engagement and constructive input have been invaluable, and we truly appreciate your time and effort in supporting this process.
>
> Best regards,
>
> Authors

---

### Official Review · Reviewer_ZUw7 · 2024-11-04

**Soundness:** 1
**Presentation:** 1
**Contribution:** 2
**Rating:** 3
**Confidence:** 3

**Summary:**

The authors propose a new maximum entropy reinforcement learning algorithm where the entropy of the next state is enforced while learning the policy. First a particular policy parameterization is used. Inner actions are first sampled according to a parameterized inner policy (i.e., a parameterized distribution from states to features, called inner actions) and the actions are transformations of these inner actions (piecewise linear in practice, such that the density of actions can be computed based on the density of inner actions using the change of variable theorem). Second, the entropy of next states is decomposed as the sum of: the entropy of the inner policy, the expected probability of the inner actions knowing the state transitions (i.e., knowing the current state and the future state), and a constant term. Then the inner policy is maximized using SAC (applying the outer actions in the MDP). The piecewise-linear transformation is computed to maximize the expectation of the probability of inner actions knowing the state transitions. The probability of (inner) actions knowing the state transition is learned by maximum likelihood estimation. This approach eventually leads to better control policies compared to algorithms that only accounts for the entropy of actions.

**Strengths:**

1. The problem at hand is very important to the RL community.
2. The approach is novel, the authors introduce a new promising intrinsic reward bonus.

**Weaknesses:**

1. Some points are unclear and have raised questions, some of which may be critical to the paper. See questions bellow.
2. The authors have missed a large part of the literature that is active in maximizing the entropy of states visited by a policy along trajectories [1, 2, 3, 4, 5]. The latter can be performed with simpler algorithms compared to the one proposed in the paper. In practice those algorithms allow to have a good state coverage, which is the objective pursued by the authors. They should be added in the related works, discussions and experiments.
3. There are errors (or shortcuts) in some equations and notations that make the paper hard to follow and prevent ensuring the correctness of all mathematical developments. Here are those I noticed:

a. In section 3, the reward function is sometimes a function of the action, sometimes not.

b. In equation (2), the distribution $P^\pi$ is undefined.

c. In section 4.1, how are $p(x)$ and $\pi$ related? (There is also a clash of notation between the constant $\pi$ and the policy $\pi$)

d. The inverse dynamic of inner policy is not defined in the main body.

e. In equation (9), I suppose an expectation is missing over the random variable $s$ in the gap term.

f. In equation (10) and (12), the variable $s$ is again undefined in the optimization problem. Is it on expectation or for all $s$, how is it done in practice?

g. Same problem in equation (13), where a function independent of $s$ equals a function of $s$.

h. In section 5.3, is the $x$-variable in the equation the inner action $e$?

i. In many equations $e$ appear as a variable, but should be replaced by $f^{-1}(a, \theta)$ as the expectations are over $a$.

j. There are thee notations for parametric functions that are used together. For example, we have $f(e, \theta)$, $f^\theta$ and $f_\theta$.

4. Section 3 focusses on defining conditions under which the action entropy equals the state entropy. The latter is done based on non-redundant actions and non-redundant policies. From my understanding, the inner policy is not non-redundant, and there are no guarantee that the (outer) policy is eventually non-redundant after optimization. While it can be argued that the discussion is in itself interesting, I think it is confusing to introduce at the very beginning of the paper something that is unused afterwards.
5. There is a methodological error in the experiment. The entropy of the next state is never shown, there is thus no evidence that the method learns high entropy policies.

[1] Lee, L., Eysenbach, B., Parisotto, E., Xing, E., Levine, S., & Salakhutdinov, R. (2019). Efficient exploration via state marginal matching. arXiv preprint arXiv:1906.05274.

[2] Guo, Z. D., Azar, M. G., Saade, A., Thakoor, S., Piot, B., Pires, B. A., ... & Munos, R. (2021). Geometric entropic exploration. arXiv preprint arXiv:2101.02055.

[3] Islam, R., Ahmed, Z., & Precup, D. (2019). Marginalized state distribution entropy regularization in policy optimization. arXiv preprint arXiv:1912.05128.

[4] Hazan, E., Kakade, S., Singh, K., & Van Soest, A. (2019, May). Provably efficient maximum entropy exploration. In International Conference on Machine Learning (pp. 2681-2691). PMLR.

[5] Liu, H., & Abbeel, P. (2021). Behavior from the void: Unsupervised active pre-training. Advances in Neural Information Processing Systems, 34, 18459-18473.

**Questions:**

1. What is the advantage of using the function $f$ to increase the gap (and thus control the next state entropy), compared to simply use as intrinsic reward the log likelihood of the inverse dynamics model (and choose f as an identity function, such that : actions = inner actions) in SAC? Similarily, why not simply learning a forward model of the MDP, and using the log likelihood of that model as intrinsic reward, to enforce the entropy of next states?
2. Could the authors clarify the different parametric functions at hand? What is the advantage of the custom transformation in section 5.3 instead of a normalizing flow?
3. A discretized multinomial distribution is used for the inverse dynamics model. What is the justification for that instead of a normalizing flow (or auto-encoder + ELBO for learning) and how is it limiting in practice?

---

> ### Author Response · Authors · 2024-11-22
> **Response to Reviewer ZUw7**
>
> Dear Reviewer,
>
> Thanks for your valuable comments.
> We hope the following statement can address your concern.
>
> **W2: Discussion with prior works focused on maximizing the entropy of all visited states**
>
> **A for W2:**
> We will add discussions of these works in the related works section and explicitly clarify the differences in objectives and methodologies between these studies and our approach.
>
> Here are the **key differences** between our method (maximizing next-state entropy) and the state entropy maximizing methods mentioned:
>
> The works mentioned in [1, 2, 3, 4, 5] directly construct policies (which can even be deterministic) to achieve better state coverage.
> However, they do not consider the entropy of policies.
>
> In contrast, our approach builds on the entropy of stochastic policies, which not only accelerates learning and prevents premature convergence to suboptimal solutions but also induces a smoother objective that connects solutions and enable the use of larger learning rates[6].
>
> Next-state entropy is an extension of policy entropy in environments with redundant action spaces.
> Policy entropy encourages action diversity while next-state entropy extends this idea to measure the diversity of effects caused by actions.
> Our method inherits the benefits of policy entropy while overcoming inefficiencies caused by action redundancy.
>
> **W3: Errors in some equations and notations:**
>
> **A for W3:**
>
> We have corrected the errors in the equations and notations and provided clarifications to address any areas of potential confusion:
>
> * a:
> Thanks for pointing out this, we have modified the notation to ensure consistency
> throughout the paper.
>
> * b: $P^{\pi}(s' \mid s)$ represents the probability of transitioning to the next state $s'$ given the current state $s$ under the policy $\pi$.
> The relationship between these two quantities can be expressed through the Bellman equation. Given a policy $\pi$, the relationship between the state transition probability $P^{\pi}(s' \mid s)$ and the action-selection probability $\pi(a \mid s)$ is as follows:
>
> $$
> P^{\pi}(s' \mid s) = \int_{a\in \mathcal{A}} \pi(a \mid s) P(s' \mid s, a)
> $$
>
> * c: To avoid confusion with the constant $\pi$, we use $\pi\_{policy}$ to represent the policy. The policy follows a normal distribution, and we modify the expression as follows:
>
> $$
> \pi\_{policy}(a) = \frac{1}{\sqrt{2 \pi} \sigma} \exp \left( -\frac{(a - \mu)^2}{2 \sigma^2} \right), \quad \mu < 0.
> $$
>
> * d: We have added a definition of inverse dynamics of inner policy in the paper. The inverse dynamics model $p_{\text{inv}}^{\phi}(e\mid s, s')$is a function that predicts the inner action $e$ required to transition from a current state $s$ to a next state $s^{\prime}$.
>
>
> * e,f,g: We acknowledge the missing expectation notation over the state $s$ in equations and have revised the paper to include it.
> All instances have been uniformly corrected in the revised version to address these issues.
>
> * h: Yes, the $x$-variable in the equation corresponds to the inner action $e$ . We map the inner action $e $ to the action $a$  using the piecewise linear function $f$.
>
> * i,j: We have standardized the notation throughout the paper to avoid any confusion.
>
> **W4: Discussion on the gap between action entropy and the next-state entropy is unused afterwards.**
>
> **A for W4:** As you pointed out, our discussion on the gap between action entropy and next-state entropy theoretically derives the condition under which non-redundant policies (and action spaces) would make them strictly equivalent. However, this condition is very strict and difficult to satisfy in cases of stochastic transitions.
>
> This discussion motivates us to propose our approach in Section 5.
> Instead of pursuing strict equivalence, we quantify the gap between action entropy and next-state entropy using a parameterized action mapping layer. We then use optimization to gradually minimize this gap, providing a more practical solution.

---

> ### Author Response · Authors · 2024-11-22
> **Response to Reviewer ZUw7 (2)**
>
> **W5: The entropy of the next state is never shown.**
>
> **A for W5:** We use a toy example similar to Section 4 where we calculate the next-state entropy for both MNSE and SAC methods. The table below shows the next-state entropy across different steps (from 10k to 150k).
>
> From the table, it is clear that both MNSE and SAC show an initial rise in next-state entropy (NSE), which promotes exploration. After the model converges to an optimal solution, the NSE decreases, reinforcing exploitation. Our method (MNSE) quickly reaches a high next-state entropy, demonstrating stronger exploration capabilities than SAC, and is able to converge more quickly to the optimal solution.
>
>
> | Step   | MNSE         | SAC          |
> |--------|--------------------|--------------------|
> | 10k    | 0.5258             | 0.5258             |
> | 20k    | 1.9093             | 1.0361             |
> | 30k    | 2.4005             | 1.2421             |
> | 40k    | 2.5332             | 1.4249             |
> | 50k    | 2.3982             | 1.5890             |
> | 60k    | 1.3882             | 1.6652             |
> | 70k    | 0.7042 (converge)  | 1.6747             |
> | 80k    | 0.0326             | 1.4930             |
> | 90k    | 0.1232             | 1.4582             |
> | 100k   | 0.1492             | 1.3715             |
> | 110k   | 0.0722             | 1.0625             |
> | 120k   | 0.1111             | 0.7042 (converge)  |
> | 130k   | 0.0072             | 0.0722             |
> | 140k   | 0.0078             | 0.1111             |
> | 150k   | 0.0525             | 0.0490             |
>
> Table 1. Next-state entropy of MNSE and SAC.
>
>
> **Q1.1: Why not simply use the log likelihood of the inverse dynamics model as intrinsic reward?**
>
> **A for Q1.1:** MinRed method [7] uses the log likelihood of the inverse dynamics model as intrinsic reward and we use it as a baseline in our paper.
> However, we demonstrate that our approach outperforms MinRed as Fig. 3 shown.
> The key advantage of our method lies in introducing an action mapping layer after the inner action, which directly shapes the action space.
> In contrast to MinRed, which uses additional intrinsic reward to train the agent, our forward approach is more direct and effective.
>
> **Q1.2: Why not simply learning a forward model of the MDP, and using the log likelihood of that model as intrinsic reward?**
>
> **A for Q1.2:** In high-dimensional state spaces, directly learning a forward model of the MDP is extremely challenging. In comparison to the state space, the action space is relatively smaller, making the training and application of inverse dynamics models more common and feasible, as seen in prior works [8, 9].
>
>
> **Q2 \& Q3: What is the advantage of using the piecewise linear function in our method rather than a normalizing flow as action transformation and What is the advantage for using the discretized multinomial distribution rather than a normalizing flow (or auto-encoder + ELBO for learning) as the inverse dynamics model?**
>
> **A for Q2 \& Q3:**
> In our theoretical derivation, any invertible mapping can serve as the action mapping function.
> On the other hand, the inverse dynamics model can be estimated using various approaches.
> In our experiments, the piecewise linear function and the discretized multinomial distribution have demonstrated performance surpassing the baseline.
> We appreciate the reviewer’s suggestion and plan to explore the integration of normalizing flows and auto-encoder + ELBO in future work to tackle more complex problems.
>
> [1] Lee, L., Eysenbach, B., Parisotto, E., Xing, E., Levine, S., & Salakhutdinov, R. (2019). Efficient exploration via state marginal matching. arXiv preprint arXiv:1906.05274.
>
> [2] Guo, Z. D., Azar, M. G., Saade, A., Thakoor, S., Piot, B., Pires, B. A., ... & Munos, R. (2021). Geometric entropic exploration. arXiv preprint arXiv:2101.02055.
>
> [3] Islam, R., Ahmed, Z., & Precup, D. (2019). Marginalized state distribution entropy regularization in policy optimization. arXiv preprint arXiv:1912.05128.
>
> [4] Hazan, E., Kakade, S., Singh, K., & Van Soest, A. (2019, May). Provably efficient maximum entropy exploration. In International Conference on Machine Learning (pp. 2681-2691). PMLR.
>
> [5] Liu, H., & Abbeel, P. (2021). Behavior from the void: Unsupervised active pre-training. Advances in Neural Information Processing Systems, 34, 18459-18473.
>
> [6]  Understanding the impact of entropy on policy optimization[C]//International conference on machine learning. PMLR, 2019: 151-160.
>
> [7]  Action redundancy in reinforcement learning[C]//Uncertainty in Artificial Intelligence. PMLR, 2021: 376-385.
>
> [8] Estimating q (s, s’) with deep deterministic dynamics gradients[C]//International Conference on Machine Learning. PMLR, 2020: 2825-2835.
>
> [9] Learning action representations for reinforcement learning[C]//International conference on machine learning. PMLR, 2019: 941-950.

---

> ### Author Response · Authors · 2024-11-25
> **Looking forward to further comments!**
>
> Dear Reviewer,
>
> We have added additional explanations and experiments for our methods. We are wondering if our response and revision have cleared your concerns. We would appreciate it if you could kindly let us know whether you have any other questions. We are looking forward to comments that can further improve our current manuscript. Thanks!
>
> Best regards,
>
> The Authors

---

> > ### Comment · Reviewer_ZUw7 · 2024-11-26
> > **Additional remarks**
> >
> > Dear authors, tank you for responding and updating the manuscript. I still have several remarks and questions that I would appreciate if you clarified.
> >
> > 1. Thank you for including papers about exploration with the marginal state entropy. I believe that you still misinterpret some of this literature. First, while most paper indeed focus on pretraining, state exploration can also be applied on-line when learning. Second, I am not convinced about the argument used in your response to justify that your approach is much different in phylosophy. I still think that an algorithm pursuing this exploration objective would strengthen your experimental setting.
> > 2. There are still some equations that may contain mistakes:
> >
> > a. In equation (9), I think that $s$, $s'$ and $e$ are undefined.
> >
> > b. In equation (10), $s_t$ is undefined and should probably be taken under the expectation.
> >
> > c. In equation (12), the relation between $a$ and $e$ is undefined.
> >
> > d. I am a bit confused why equation (11) is written as a sum and the others are not.
> >
> > 3. The article would benefit to include a proper discussion and illustration of the next state entropy.
> > 4. It is unfair to claim (in your response) that your approach is mode-free when you learn the inverse dynamics.

---

> > > ### Author Response · Authors · 2024-11-27
> > > **Response to Reviewer ZUw7 (1/2)**
> > >
> > > Dear Reviewer,
> > >
> > > Thanks for your reply! We will address your follow-up questions below.
> > >
> > > **Q1.1: Misinterpretation of exploration with marginal state entropy:**
> > >
> > > **A for Q1.1:** Thank you for your valuable feedback. We have revised our statement in the related work section in the revised paper to better reflect the literature. The corrected description is as follows:
> > >
> > > State Entropy Maximization aims to learn a reward-free policy in which state visitations are uniformly distributed across the state space, thus promoting robust policy initialization and efficient adaptation.
> > > **Additionally, when task rewards are available, incorporating state entropy as an intrinsic reward has proven to be an effective approach for enhancing exploration.**
> > >
> > > **Q1.2: Pursuing the state entropy maximization objective would strengthen the experimental setting.**
> > >
> > > **A for Q1.2:**
> > > As suggested, we conducted additional experiments on MuJoCo tasks with input nonlinearity, comparing our method with APT [1] and SMM [2].
> > > The hyperparameters are consistent with the defaults in URLB [9].
> > > As shown in the table below, our method outperforms the baseline approaches.
> > >
> > > The key advantage of our method lies in the introduction of an action mapping layer following the inner action, which directly shapes the action space. In contrast to APT and SMM, which rely on additional intrinsic rewards to train the agent, our approach is more direct and effective.
> > >
> > > |Method| Ant | HalfCheetah | Hopper | Walker2d |
> > > |--------------------|--------------------|--------------------|--------------------|--------------------|
> > > | APT[1] | 1985.99±655.18 | 7199.61±1784.04 | 2559.85±559.68 | 2584.04±739.52 |
> > > | SMM[2] | 1550.26±419.62 | 7275.25±1477.57 | 1920.11±370.66 | 2927.85±975.06 |
> > > | MNSE (Ours) | **3586.25±681.38** | **7824.08±1190.98** | **3013.21±149.09** | **4240.30±484.45** |
> > >
> > > Table 3. Comparisons with additional baselines in MuJoCo tasks with input nonlinearity.
> > >
> > > **Q2: Errors in some equations and notations:**
> > >
> > > **A for Q2:**
> > > Thanks for your detailed and valuable suggestions.
> > > We have corrected the errors in the equations and provided clarifications to address any areas of potential confusion:
> > >
> > > * a: We have made clearer annotations in equation (9), replacing $s$ with $s\_t$ and $s^{\prime}$ with $s\_{t+1}$, and we have labeled $e = f^{-1}(a\_t; \theta)$.
> > >
> > > * b: In equation (10), we have updated the expectation notation to $\underset{\substack{(s\_t, a\_t, s\_{t+1}) \sim \pi\\ s\_{t+1} \sim P(\cdot \mid s\_t, a\_t)}}{\mathbb{E}}$ for consistency and to avoid confusion.
> > >
> > > * c: In equation (12), we explicitly define the relationship between $a$ and $e$ as $a = f(e; \theta)$.
> > >
> > > * d: In equation (11), to ensure consistency, we have changed the sum symbol to the expectation symbol.
> > >
> > > **Q3: Include a proper discussion and illustration of the next state entropy.**
> > >
> > > **A for Q3:**
> > > Thank you for your suggestion. We have added a "Discussion" section (line 517) in the revised paper to elaborate on why maximizing next-state entropy is important in reinforcement learning. The content is as follows:
> > >
> > > Why Maximize Next-State Entropy in Reinforcement Learning?
> > > Entropy regularization is a fundamental technique in reinforcement learning.
> > > By integrating an entropy maximization term,
> > > it enhances robustness to model and estimation errors [3],
> > > promotes the acquisition of diverse behaviors [4],
> > > facilitates broader exploration [5,6,7]
> > > and accelerates the learning process by smoothing the optimization landscape [8].
> > >
> > > However, maximizing policy entropy may not directly promote policy optimization due to redundancy in the action space. In such cases, next-state entropy extends the concept of policy entropy more directly. Specifically, next-state entropy measures the entropy of the next state resulting from the policy, rather than the action itself. This shift allows next-state entropy to capture the diversity of effects induced by actions. By bridging the gap between next-state and policy entropy, our method retains the benefits of policy entropy while addressing inefficiencies caused by action redundancy.
> > >
> > > **Q4: Claim the approach is mode-free.**
> > >
> > > **A for Q4:**
> > > Thanks for your suggestions and we have removed this point in our response.
> > >
> > > Best,
> > >
> > > The Authors

---

> > > ### Author Response · Authors · 2024-11-27
> > > **Response to Reviewer ZUw7 (2/2)**
> > >
> > > [1] Liu, H., & Abbeel, P. (2021). Behavior from the void: Unsupervised active pre-training. Advances in Neural Information Processing Systems, 34, 18459-18473.
> > >
> > > [2] Lee, L., Eysenbach, B., Parisotto, E., Xing, E., Levine, S., & Salakhutdinov, R. (2019). Efficient exploration via state marginal matching. arXiv preprint arXiv:1906.05274.
> > >
> > >
> > > [3] Ziebart B D. Modeling purposeful adaptive behavior with the principle of maximum causal entropy[M]. Carnegie Mellon University, 2010.
> > >
> > > [4] Haarnoja T, Tang H, Abbeel P, et al. Reinforcement learning with deep energy-based policies[C]//International conference on machine learning. PMLR, 2017: 1352-1361.
> > >
> > > [5] Fox R, Pakman A, Tishby N. Taming the noise in reinforcement learning via soft updates[J]. arXiv preprint arXiv:1512.08562, 2015.
> > >
> > > [6] Haarnoja T, Zhou A, Abbeel P, et al. Soft actor-critic: Off-policy maximum entropy deep reinforcement learning with a stochastic actor[C]//International conference on machine learning. PMLR, 2018: 1861-1870.
> > >
> > > [7] Haarnoja T, Zhou A, Hartikainen K, et al. Soft actor-critic algorithms and applications[J]. arXiv preprint arXiv:1812.05905, 2018.
> > >
> > > [8] Ahmed Z, Le Roux N, Norouzi M, et al. Understanding the impact of entropy on policy optimization[C]//International conference on machine learning. PMLR, 2019: 151-160.
> > >
> > > [9] Laskin M, Yarats D, Liu H, et al. Urlb: Unsupervised reinforcement learning benchmark[J]. arXiv preprint arXiv:2110.15191, 2021.

---

> ### Author Response · Authors · 2024-12-02
>
> Dear Reviewer,
>
> Thank you for your thoughtful feedback on our paper. With only two days remaining in the discussion period, we kindly ask that you review our responses to ensure we have fully addressed your concerns. If you find our responses satisfactory, we would greatly appreciate it if you could reconsider your rating/scoring.
>
> Your engagement and constructive input have been invaluable, and we truly appreciate your time and effort in supporting this process.
>
> Best regards,
>
> Authors

---

### Official Review · Reviewer_S96H · 2024-11-05

**Soundness:** 3
**Presentation:** 4
**Contribution:** 2
**Rating:** 6
**Confidence:** 3

**Summary:**

This article presents a new reinforcement learning method called Maximum Next State Entropy (MNSE) which optimizes next-state entropy through a reversible action mapping layer. MNSE shows better performance than existing methods in complex environments with nonlinear actuators and emphasizes the importance of appropriate model and parameter settings.

**Strengths:**

1. The experiments cover multiple continuous control tasks, including complex environments like robotic arm control. The results show that MNSE outperforms traditional maximum entropy reinforcement learning methods and other reward-based exploration strategies in these tasks. This indicates the significant potential of the MNSE method in practical applications.


2. The paper provides rigorous theoretical analysis and demonstrates its effectiveness, which is significant for advancing research and development in the field of reinforcement learning.


3. The paper is written in a clear and concise manner. This helps readers better understand and grasp the core ideas and technical features of the method.

**Weaknesses:**

Given that MNSE relies on the accurate estimation of the dynamic model, how do you ensure the accuracy of these estimations and avoid overfitting?

Additionally, could you provide guidance on how to reasonably select the hyper-parameters to optimize the algorithm's performance?

**Questions:**

See questions in Weaknesses.

---

> ### Author Response · Authors · 2024-11-22
> **Response to Reviewer S96H**
>
> Dear Reviewer,
>
> Thanks for finding our work is well written and significant for advancing research.
> The points you raised are explained in the following.
>
> **W1: How do you ensure the accuracy of the dynamic model estimations in MNSE and avoid overfitting?**
>
> **A for W1:**
>
> We ensure the accuracy of the dynamic model estimations in the following ways:
>
> - **Choice of distribution:**
>    We adopt discrete multinomial distributions rather than Gaussian distributions for the inverse dynamic model. This choice better captures the dynamic characteristics of real-world physical systems.
>
> - **Iterative updates:**
>    The inverse dynamic model is continuously updated throughout the whole training process. This iterative refinement ensures that the model adapts and maintains accuracy as training progresses.
>
> To avoid overfitting, we employ the following strategies:
>
> - **Continuously collected data:**
>    During the iterative training of the dynamic model, the replay buffer $D$  continuously incorporates new samples collected by the policy. This ongoing data augmentation enhances diversity and improves the model's generalization ability, effectively preventing overfitting.
>
> - **Regularization:**
>    We include a weight decay term in the Adam optimizer to constrain the L2-norm of the model parameters, further mitigating overfitting risks.
>
>
> **W2: Could you provide guidance on how to reasonably select the hyperparameters to optimize the algorithm's performance?**
>
> **A for W2:**
>
> - **Selection of hyperparameters in SAC backbone:**
> Our algorithm, MNSE, is developed based on the SAC algorithm from the RL Baselines3 Zoo. All hyperparameters (e.g., learning rate, buffer size) are consistent with the SAC defaults in RL Baselines3 Zoo, as these have already been optimized. This ensures a fair comparison with baseline methods.
>
> - **Selection of unique parameters in MNSE:**
> For our algorithm, a key hyperparameter is $N$, the number of parameters in the piecewise linear function. As illustrated in Figure 5, the algorithm's performance improves as $N$ increases. However, beyond $N \geq 20$, the performance stabilizes. For control tasks similar to Mujoco or Metaworld, we recommend setting $N=20$.

---

> ### Author Response · Authors · 2024-11-25
> **Looking forward to further comments!**
>
> Dear Reviewer,
>
> We have added additional explanations for our methods. We are wondering if our response and revision have cleared your concerns. We would appreciate it if you could kindly let us know whether you have any other questions. We are looking forward to comments that can further improve our current manuscript. Thanks!
>
> Best regards,
>
> The Authors

---

> ### Author Response · Authors · 2024-12-02
>
> Dear Reviewer,
>
> Thank you for your thoughtful feedback on our paper. With only two days remaining in the discussion period, we kindly ask that you review our responses to ensure we have fully addressed your concerns. If you find our responses satisfactory, we would greatly appreciate it if you could reconsider your rating/scoring.
>
> Your engagement and constructive input have been invaluable, and we truly appreciate your time and effort in supporting this process.
>
> Best regards,
>
> Authors

---

### Note · Authors · 2024-12-09

I have read and agree with the venue's withdrawal policy on behalf of myself and my co-authors.